# Spatial and temporal localization of immune transcripts defines hallmarks and diversity in the tuberculosis granuloma

Berit Carow [1,4], Thomas Hauling[2,4], Xiaoyan Qian [2], Igor Kramnik[3], Mats Nilsson[2] & Martin E. Rottenberg[1]

Granulomas are the pathological hallmark of tuberculosis (TB) and the niche where bacilli can grow and disseminate or the immunological microenvironment in which host cells interact to prevent bacterial dissemination. Here we show 34 immune transcripts align to the morphology of lung sections from *Mycobacterium tuberculosis*-infected mice at cellular resolution. Colocalizing transcript networks at <10 μm in C57BL/6 mouse granulomas increase complexity with time after infection. B-cell clusters develop late after infection. Transcripts from activated macrophages are enriched at subcellular distances from *M. tuberculosis*. Encapsulated C3HeB/FeJ granulomas show necrotic centers with transcripts associated with immunosuppression (*Foxp3*, *Il10*), whereas those in the granuloma rims associate with activated T cells and macrophages. We see highly diverse networks with common interactors in similar lesions. Different immune landscapes of *M. tuberculosis* granulomas depending on the time after infection, the histopathological features of the lesion, and the proximity to bacteria are here defined.

[1] Department of Microbiology, Tumor and Cell Biology, Karolinska Institutet, 171 77 Stockholm, Sweden. [2] Science for Life Laboratory, Department of Biochemistry and Biophysics, Stockholm University, 171 65 Solna, Sweden. [3] Department of Medicine, Boston University School of Medicine, Boston, Massachusetts 02118, USA. [4]These authors contributed equally: Berit Carow, Thomas Hauling. Correspondence and requests for materials should be addressed to M.E.R. (email: Martin.Rottenberg@ki.se)

Tuberculosis (TB), caused by infection with *Mycobacterium tuberculosis*, remains a leading public health problem worldwide. In 2016, 10.4 million people developed TB and 1.7 million patients died from the disease[1]. The challenges to control TB are enormous: resistance to available drugs against *M. tuberculosis* is increasing, whereas the only available vaccine against TB is only partially protective. One quarter of the global population is latently infected with *M. tuberculosis* with a 10% probability of those infected to develop active TB during their lifetime. To understand why reactivation of *M. tuberculosis* occurs in some but not in other individuals is central for development of new vaccines and therapies.

Infection occurs when the inhaled *M. tuberculosis* are phagocytized by resident lung alveolar macrophages[2]. Infected cells recruit mononuclear phagocytes to the infection site, forming a nascent granuloma. During the subclinical stage of infection, the granuloma provides the immune environment required for the containment of bacteria. *M. tuberculosis*-specific T cells are crucial for the granuloma maturation, maintenance, and control of the bacterial spread[3]. However, if due to impaired immunity the integrity of the granuloma is lost, reactivation of *M. tuberculosis* leads to the destruction of the lung structure and to the transmission of *M. tuberculosis* to other humans.

In association with the diverse outcome of infection, studies from autopsies have shown important diversity in the granuloma histology. In addition to the encapsulated granuloma with a caseous necrotic center, TB granulomas can be non-necrotizing, neutrophil-rich, mineralized, fibrotic, or cavitary[4,5]. TB granulomas in commonly used inbred mouse strains such as C57BL/6 do not develop necrotizing lesions, whereas encapsulated necrotizing granulomas are found in other strains such as the C3HeB/FeJ[6–8]. Although the histological features of granulomas have been well characterized, the immune and inflammatory mechanisms that underlie variable granuloma dynamics and clinical outcomes of TB infection remain to be further elucidated.

In order to understand the immunological architecture of the murine TB granulomas, we have used a method for highly sensitive multiplexed in situ imaging of selected immune messenger RNA species. The method, in situ sequencing, based on rolling-circle amplification (RCA) of padlock probes and on sequencing-by-ligation chemistry produced highly specific amplified products that enabled detection of individual mRNA molecules in situ in the unperturbed context of fixed tissues at cellular resolution[9,10].

Here, using in situ sequencing, the topography *M. tuberculosis* granulomas were compared in lungs from mice at different times after infection and from mice developing encapsulated or non-encapsulated granulomas. Colocalizing transcript networks at <10 μm in lesions in the same lung with similar or different histological characteristics were contrasted. Moreover, networks of co-expressed transcripts in different areas within lesions were defined.

## Results

### Specificity, reproducibility, and performance of in situ sequencing.
The in situ sequencing technique was used to localize simultaneously 34 immune transcripts immune in paraformaldehyde-fixed sections of lungs from *M. tuberculosis*-infected mice. Transcripts coding for chemokine receptors, cytokines, effector molecules, transcription factors, and surface molecules that define immune cell populations were targeted. These molecules play a major role in the cellular immune control of the *M. tuberculosis* infection. Three consecutive sections from C57BL/6 mice at 3, 8, and 12 weeks after aerosol infection with *M. tuberculosis* (wpi) and those from C3HeB/FeJ mice that develop encapsulated necrotic granulomas were used[11].

Transcripts were aligned with the histopathological features of the same lung section (Supplementary Fig. 1).

Nonspecific signals (when base calling did not correspond to those built in the barcoding sequences) were minimized by increasing the signal threshold, whereas the density of specific signals remained mostly unaltered, indicating a high specificity of the reaction (Supplementary Fig. 2A). At the fixed threshold selected (0.45), the performance (the total number of signals) ranged from 7 to $15 \times 10^4$ signals per section (Supplementary Fig. 2B). All transcripts were simultaneously and differentially detected in the lungs (Supplementary Fig. 2C). As expected, *Cc10* mRNA, expressed by Clara cells, located in the epithelium of pulmonary airways, and inducible nitric oxide synthase *(Inos)*, expressed by inflammatory cells, localized in the granuloma. These showed similar distribution in consecutive slides (Supplementary Fig. 2D). The ratio of the density of specific mRNAs in annotated lesions vs. unaffected areas (defined in Supplementary Fig. 1) from consecutive sections were similar for some but the variance was higher for other transcripts (Supplementary Fig. 2E). This usually related to the higher sparsity of signals in the latter group. If the mRNA species was undetectable in more than two areas, it was excluded from further analysis.

### Maturation of the granuloma.
Most immune transcripts localized within the granulomas as shown for *Cd68*, *Inos*, and *Cd3e* mRNA (Fig. 1a). Statistical analysis of extracted reads at 3 wpi showed increased localization of all but four sparsely detected mRNA species (*Cd4*, *Cd40l*, *Elane*, and *Il17a*) in the lesions compared with non-affected area. All immune transcripts except *Cd4* and *Cd40l* preferentially located in the C57BL/6 granuloma at 12 wpi ($p < 0.01$, $\chi^2$-test).

The density of *Actb* mRNA, coding for β-actin, was two- to threefold increased in granulomas compared with the unaffected regions, reflecting higher cellular density in the lesions (Fig. 1b and Supplementary Fig. 2E).

Several transcripts commonly expressed by myeloid cells (i.e., *Tnf*, *Inos*, and *Il6*) showed similar relative frequencies in granulomas from lungs at 3 and 12 wpi studied as compared with unaffected areas (Fig. 1b). Instead, increased relative densities were observed for several transcripts at 12 wpi, some of which associated with adaptive immune responses (i.e., *Cd19*, *Ccr6*, *Ifng*, and *Cd3e* mRNA) (Fig. 1b).

We then analyzed transcripts co-expressed at <10 μm using the Insitunet app[12]. Transcripts with significant spatial co-expression were displayed as edges in a network map. Interactions were only studied when observed in at least two consecutive slides (Fig. 2a). As an example, out of 561 possible mathematical combinations of pairs of the 34 transcripts at 8 wpi studied ($C_{34,2}$), 19 transcript pairs significantly colocalized (Supplementary Table 1).

We observed that granulomas showed co-expression of *Inos* and different T-cell-related transcripts (*Cd3e*, *Tcrb*, *Cd8b*) at 3 wpi (Fig. 2a). *Inos* sequences were also the principal interacting node in granulomas at 8 and 12 wpi, when the number of colocalizing transcripts in the granulomas increased (Fig. 2a). No transcript co-expression was detected in unaffected regions of the sections.

### Distinct localization of transcripts within the granuloma.
An unsupervised clustering of mRNA densities across the pulmonary tissue was then performed. The tissues were divided into hexagons (hexbin) with a 70 μm-long radius (Fig. 2b). The hexbins were separated into the minimal number of clusters showing different sequence frequencies (3 clusters for 3 and 8 wpi, 4 clusters 12 wpi). The hexbin clusters identified areas that corresponded either to the granulomas or to unaffected sites (as

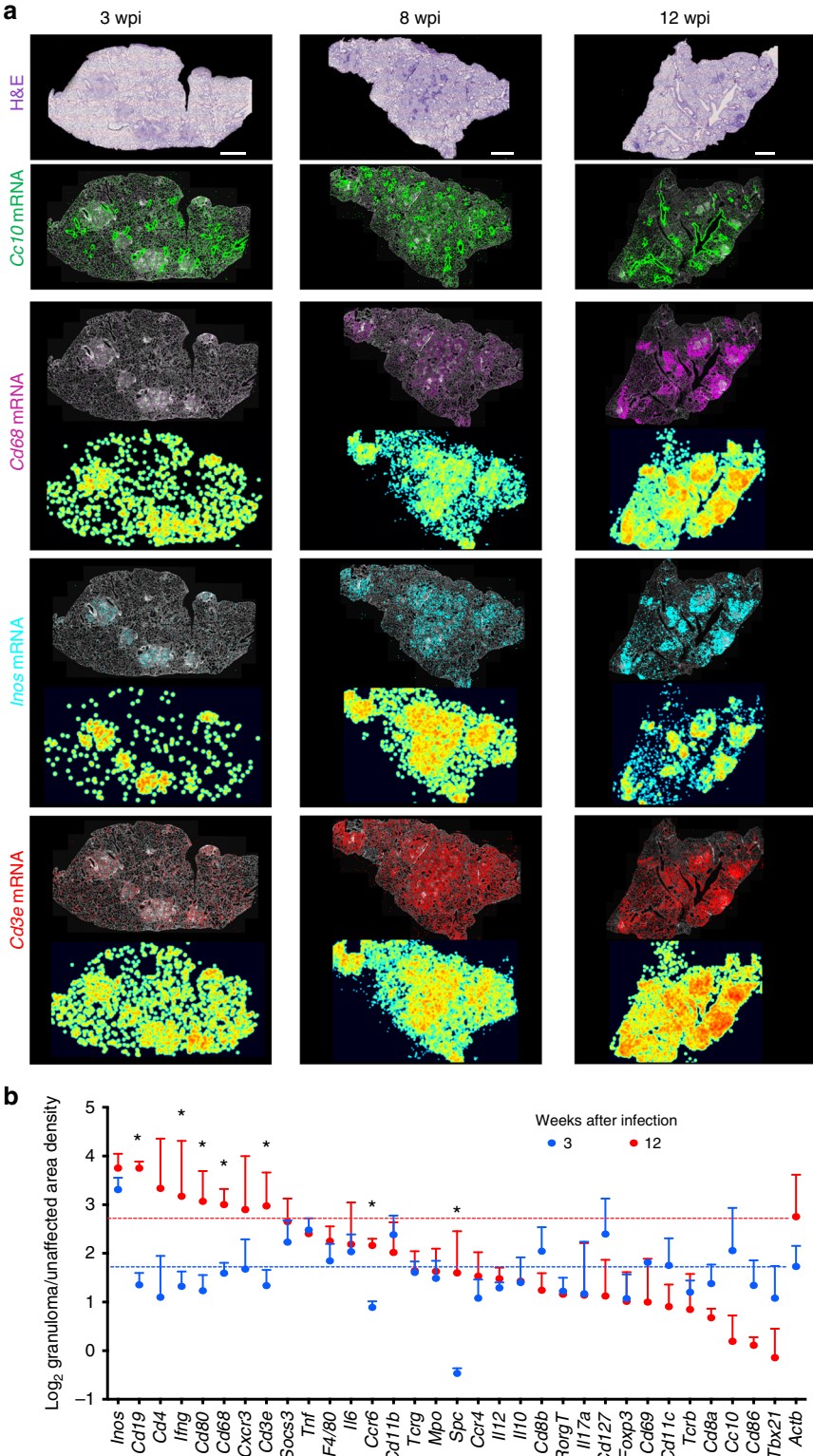

**Fig. 1** Preferential localization of immune sequences in granulomas during infection with *M. tuberculosis*. **a** At 3, 8, and 12 weeks post *M. tuberculosis* infection (wpi), the fixed lung tissue sections were stained with hematoxylin–eosin (H&E) and signals for *Cc10*, *Cd68*, *Inos*, and *Cd3e* were plotted on DAPI labeling as background. Pseudo-color density XY positional log$_2$ plots of transcript representations are shown below for *Cd68*, *Inos*, and *Cd3e* transcripts. One representative of three consecutive sections is displayed. Scale: 1000 μm. **b** The ratio of amplified transcripts in granulomas vs. unaffected regions was calculated for each transcript. The mean log$_2$ ratio of transcript density in the granuloma in relation to the density in unaffected region + SEM in three consecutive sections is depicted. Sections from lungs at 3 and 12 wpi are compared. *Differences in relative transcript density at 3 and 12 wpi are significant ($p < 0.05$ unpaired Student's *t*-test). Source data are provided as a Source Data file

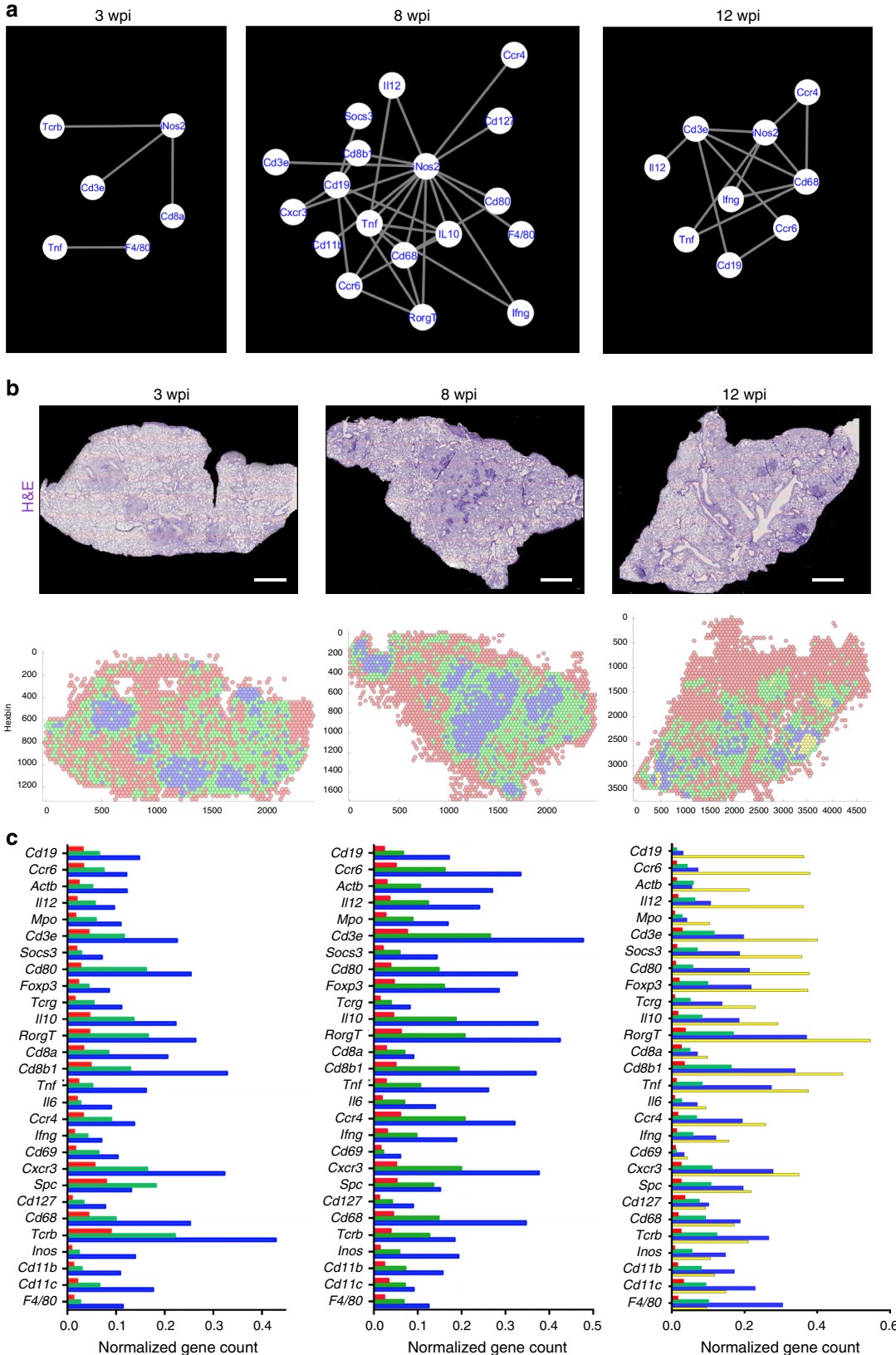

defined by hematoxylin–eosin (H&E) staining, Supplementary Fig. 1) at all times of infection (Fig. 2b). At 3 wpi, clusters differed by the concentration but not on the relative densities of transcripts (Fig. 2b, c). One cluster (blue) located in the granuloma, another (red) in unaffected areas, and the third (green) around the granuloma in areas with mild inflammation or unaffected

(Fig. 2b and Supplementary Fig. 1). In contrast, at 12 wpi the clusters contained different relative frequencies of transcripts (Fig. 2b, c). Three of the four hexbin clusters located in the granuloma area. One of these (green) also localized in areas surrounding the granuloma (Fig. 2b). Another cluster (yellow) overlapped with lymphoid-rich areas within the granuloma and

**Fig. 2** Maturation of the granuloma. **a** The spatial co-expression relationships between the in situ sequencing data were converted into network-based visualization, where each unique transcript is a node in the network and edges represent the interactions using InsituNet. To identify spatially co-expressed transcripts, InsituNet analyzed the co-occurrence of transcript detections within 30 pixels (10 μm). Euclidean distance for each pairwise combination of transcripts (see Supplementary Table 1 as example). Representative examples of one lesion per time point as defined in Supplementary Fig. 1 were selected. The significantly co-expressed sequences that are common in the same lesion from at least two consecutive sections are here depicted. *Actb* mRNA was excluded from network analysis due to broad expression in different cellular populations. **b** The tissue section plane was uniformly tiled into 200 pxls (70 μm) radius hexagons and the density of the multiple sequences in each hexagon was aggregated by binning and displayed into a 2D-hexbin map. The densities of the sequences were organized by clustering the hexagons into 3 (3 and 8 wpi) or 4 (12 wpi) different expression patterns. The H&E staining of one representation lung section and their hexbin maps at 3, 8, and 12 wpi are shown. Scale: 1000 μm. **c** The mean centroid normalized transcript counts in each hexagon was compared for the different clusters. The color-code used for the bars corresponds to that in the 2D-hexbin map. Note that the red clusters corresponding to unaffected areas contained less counts for of all specific sequences. Note also at 12 wpi that although some sequences were dominant in single clusters (i.e., *Cd19* mRNA in the yellow cluster), other sequences were more evenly distributed, or predominated in the blue clusters. Source data are provided as a Source Data file

showed an over-representation of *Cd19* mRNA, expressed by B cells, and several other transcripts (*Ccr6*, *Il12*, *Mpo*, *Cd3e*) (Fig. 2c). However, myeloid markers (*Cd68*, *Inos*, *Cd11b*, *Cd11c*) were mainly expressed in the blue cluster. Thus, distinct areas within the granulomas containing different densities of transcripts were revealed by an unbiased analysis of the images. This indicated an increased compartmentalization of granulomas at late time points after infection.

*Cd19* mRNA showed a sparse and indiscriminate expression at 3 wpi, but a strong focal distribution in lymphoid-rich areas that resembles inducible B-cell follicles[13,14] within the C57BL/6 granuloma at later time points (Fig. 3a). Immunohistochemical labeling of B cells with anti-B220 in these sections resulted in a similar pattern, further confirming the specificity of the in situ sequencing (Fig. 3b).

Epithelioid areas in 12 wpi granulomas contained a high density of *Cd68* mRNA and were devoid of *Cd19* mRNA (Fig. 3c). Instead, *Cd3e* mRNA colocalized with both *Cd68* and with *Cd19* mRNA at the epithelioid and lymphoid areas (Fig. 3c). *Inos* and *Cd68* mRNA localized in the same areas of the lesions, whereas other transcripts, such as *Tnf* and *Il12p35/40* mRNA only partially colocalized with the former transcripts (Supplementary Fig. 3A). *Ccr6* and *Cd19* mRNA showed a similar localization (Fig. 3c). Double immunolabeling of CD3 and CD68 confirmed results obtained by in situ sequencing, as CD3+ cells located both in 4′,6-diamidino-2-phenylindole (DAPI)-rich and surrounding, DAPI-lighter areas of the lesion, whereas CD68+ located in DAPI-light granuloma areas (Supplementary Fig. 3B).

The density of several transcripts including *Ccr6*, *Cd19*, and *Il12*, and to a lesser degree *Cd3e*, *Cxcr3*, and *Cd8b* mRNA was increased in the lymphoid in relation to the epithelioid areas (Fig. 3d) after a supervised annotation of the lymphoid and the epithelioid areas of the granulomas based on H&E staining. *Cc10* and myeloid-associated markers were found among the under-represented transcripts (Fig. 3d). In contrast, *Cd11b*, *Inos*, and *Cd68* mRNA were enhanced in the epithelioid as compared with the lymphoid area, whereas a third group (including *Il6*, *Tnf*, *Ifng*, *Cd127*, and *Ccr4* mRNA) showed similar densities in both areas (Fig. 3d).

Thus, in an unsupervised way or after annotation based on histological features, different areas of the granuloma were defined on the frequencies and distribution of transcripts.

Distinct networks of co-expressed transcripts in lymphoid and epithelioid areas of the granulomas were apparent. In lymphoid areas, *Cd19* mRNA was a major node interacting with *Cd3e* and *Ccr6* mRNA at 8 and 12 wpi (Fig. 4a, b). *Il12*, *Ifng*, *Tnf*, and *Cxcr3* mRNA were also found in lymphoid networks at 12 wpi (Fig. 4a). *Mpo* (neutrophil myeloperoxidase) and *Rorg* transcripts colocalized with *Cd19* mRNA as well (Fig. 4a). Thus, transcripts from T and B cells are in close proximity in the lymphoid areas.

On the other hand, *Inos*, the main interacting node in the epithelioid area, co-expressed with *Cd68* mRNA (Fig. 4b). Colocalizing sequences in some epithelioid areas were *Ccr4*, *Tnf*, and other transcripts expressed by T cells (Fig. 4b). Density plots of *Cd3e*, *Cd8b*, *Cxcr3*, and *Ifng* mRNA showed a similar and widespread location throughout the granuloma of T-cell-related transcripts that only in part overlapped with *Ccr4* mRNA (Supplementary Fig. 4A, B).

**Transcripts colocalizing with *M. tuberculosis*.** To identify transcripts located in proximity to *M. tuberculosis*, sections were stained with Auramine–Rhodamine T, scanned, and aligned to the transcript signals. The transcripts located at <3, 10, 30, 300, and 600 μm distance from *M. tuberculosis* bacteria were extracted. *Inos* and *Cd68* mRNA at <10 μm from *M. tuberculosis* are shown (Fig. 4c, d). The frequencies of transcripts localized at expanding distances from the bacteria converged with those in the whole lung, validating the results (Supplementary Fig. 5A, B and Fig. 4e). *Inos*, *Cd68*, *Cd11b*, *Tnf*, and *Socs3* mRNA were enriched at shorter distances to *M. tuberculosis*, suggesting that activated macrophages expressing these transcripts colocalize with bacteria in the granuloma (Fig. 4e). Similar results were observed in three consecutive slides analyzed as depicted for a selected number of transcripts (Supplementary Fig. 5C). On the other hand, transcripts such as *Cc10*, *Spc* (expressed by type II alveolar epithelial cells), *Il6*, and *Foxp3* showed an opposite trend: the relative frequency of these transcripts decreased at shorter distances to *M. tuberculosis* bacteria (Fig. 4e). Other transcripts including *Cd8b*, *Cd3e*, *Cxcr3*, *Ccr4*, and *Il12* showed similar frequencies at different distances from *M. tuberculosis*, suggesting a neutral spatial association with the bacteria. Interestingly, *M. tuberculosis* and adjacent transcripts located in the epithelioid areas of the granuloma, in proximity to the lymphoid areas, rather than randomly spread throughout the whole lesion (Fig. 4f).

**Heterogeneity of granulomas.** The mRNA density in granulomas from the same lung was then compared. Heat maps and principal component analysis showed that three out of four granulomas showed similar transcript compositions at 3 wpi, whereas a fourth diverged (Fig. 5a–c). The three lesions depicted at 12 wpi were heterogeneous (Fig. 5b–d). All lesions were different compared with unaffected areas (Fig. 5a–d). The ratio of transcript densities in epithelioid and lymphoid areas of different granulomas within the same lung at 12 wpi also showed substantial variation (Fig. 5e).

Qualitative differences in networks in lymphoid or epithelioid areas of granulomas from the same lung were also observed (Fig. 6a, b). Despite such variation, common networks of transcripts to all epithelioid or lymphoid area determined

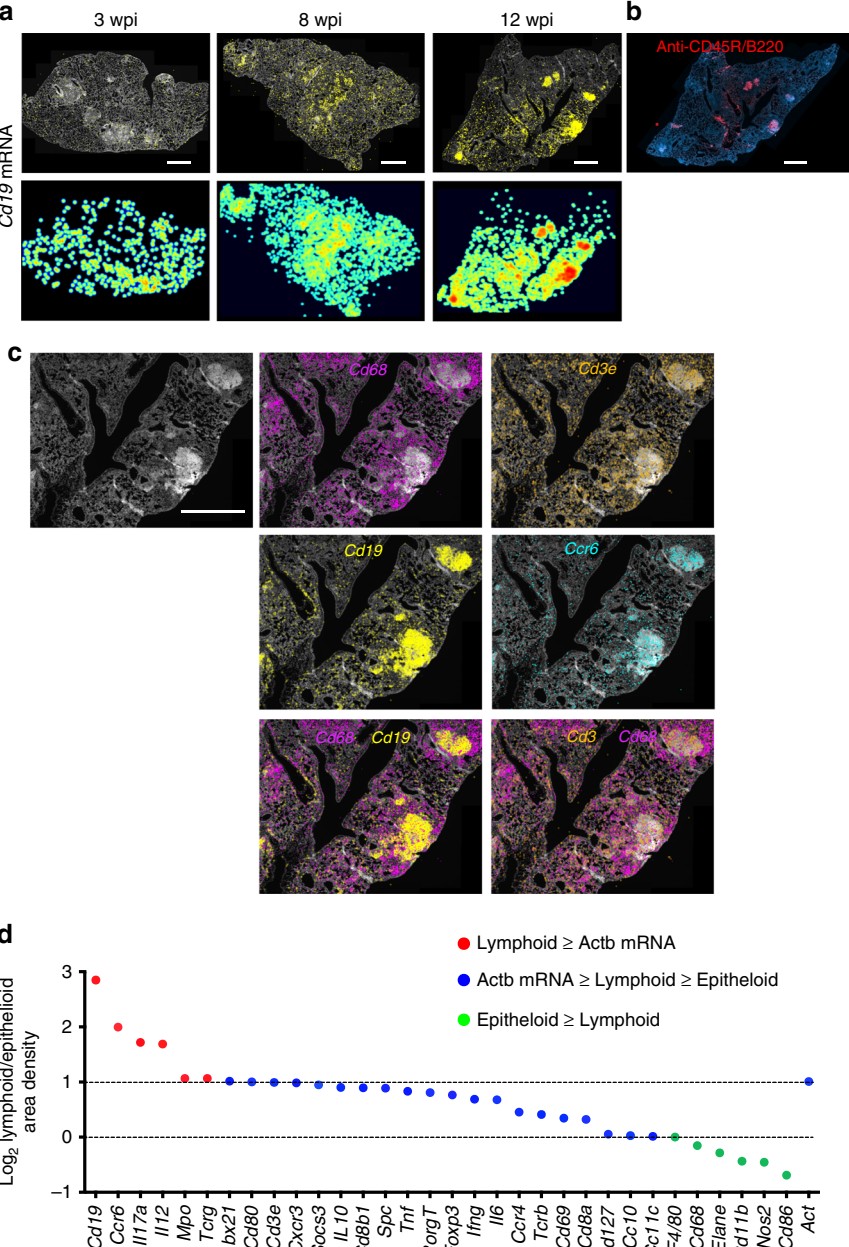

**Fig. 3** Distinct localization of *Cd19* mRNA within the lymphoid-rich areas in the granuloma. **a** In situ detection of *Cd19* mRNA transcripts in lungs from *M. tuberculosis*-infected animals. For one representative of three consecutive sections per time point, the DAPI staining, *Cd19* mRNA raw signals, and pseudo-color log$_2$ density plots are shown. Scale: 1000 μm. **b** Immunohistochemical labeling of CD45R/B220. Note the similar pattern of the labeling as compared with the in situ staining for *Cd19* mRNA in a consecutive section at 12 wpi in **a**. Scale bar: 1000 μm. **c** The expression of *Cd68*, *Cd3e*, *Cd19*, and *Ccr6* mRNA in one representative granuloma from C57BL/6-infected mice at 12 wpi is shown. *Cd68* and *Cd19* sequences locate in distinct areas of the granuloma, whereas *Cd3e* mRNA locates in both *Cd68* and *Cd19* mRNA-rich areas. Scale bar: 1000 μm. **d** The density of sequences in the epithelioid or lymphoid areas as defined in Supplementary Fig. 1 were quantified in three consecutive sections at 12 wpi. The ratios of sequence densities in lymphoid/epithelioid areas were calculated per section and the mean is depicted. The ratios were color coded accordingly if their relative frequency was higher (red) to *Actb* mRNA or if lower than the frequencies in epithelioid cells (green). Transcripts with ratios of lymphoid/epithelioid >1 and less than the ratio of *Actb* mRNA are depicted in blue. Source data are provided as a Source Data file

*Inos-Cd68* and *Cd19* mRNA as the main interacting nodes, respectively (Fig. 6c). Those differed substantially from the core networks of necrotic granulomas in the C3HeB/FeJ mice analysed below (Fig. 6d).

**Comparison of encapsulated and non-encapsulated granulomas.** Both encapsulated and non-encapsulated lesions were

detected in C3HeB/FeJ *M. tuberculosis*-infected lung sections (Fig. 7a). Highly organized encapsulated granulomas showed a center with caseous and sarcoid necrotic cells surrounded by a fibrotic capsule and a layer of foamy macrophages (Supplementary Fig. 6A). The granuloma contour showed areas with lymphoid and epithelioid cells, as previously described (Supplementary Fig. 6A). *M. tuberculosis* accumulated mainly in the rim of the necrotic granulomas rather than in the center

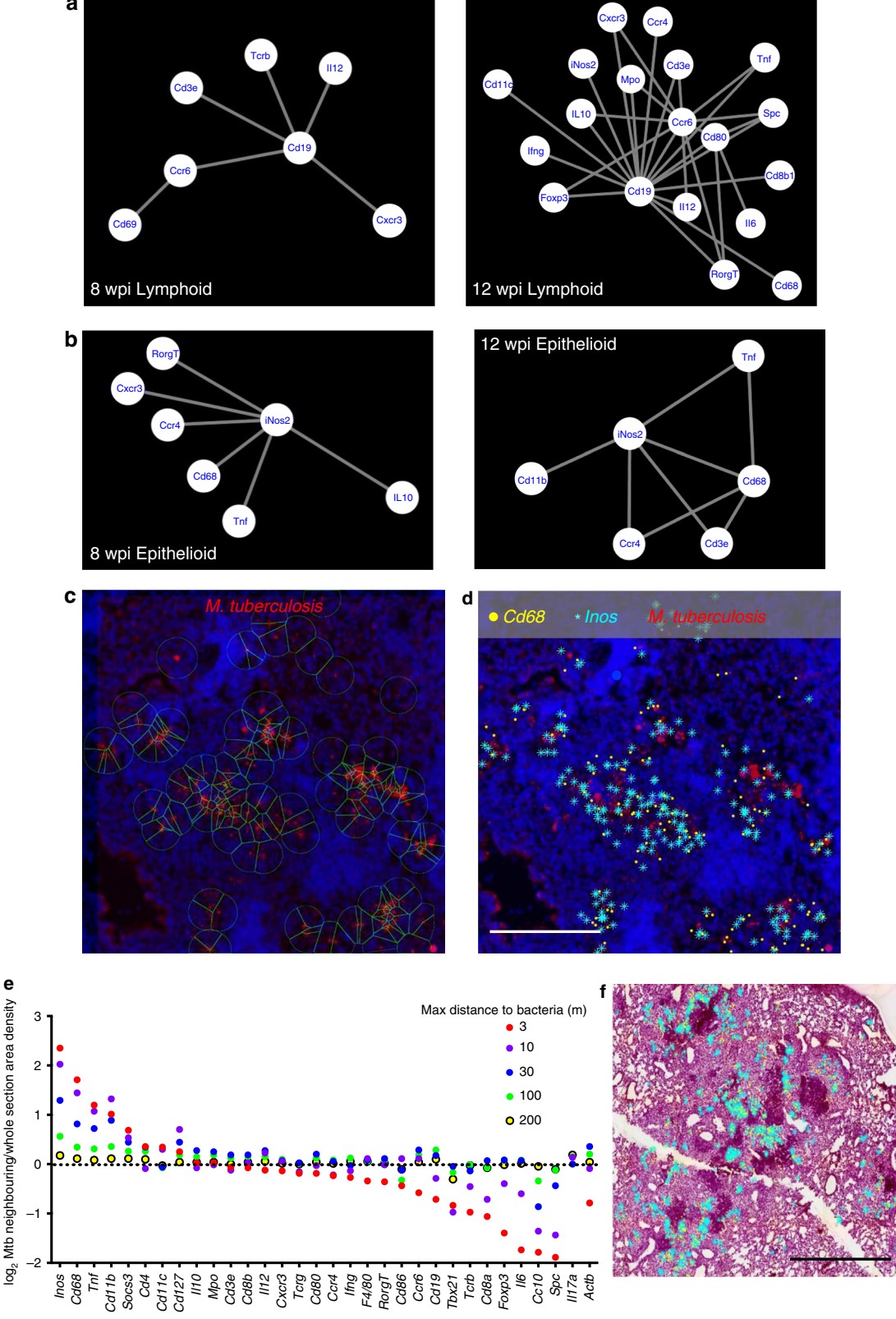

(Fig. 7b and Supplementary Fig. 6C) confirming previous data[15]. The center (GC) and the outline (GO) of seven large encapsulated granulomas, five non-organized perivascular cellular lesion with high density of lymphocytes and epithelioid cells, with pockets of intracellular bacteria (non-organized granuloma, NOG) (Fig. 7a

and Supplementary Fig. 6B, D), two smaller encapsulated granuloma with high bacterial density, areas of necrosis and neutrophilic infiltrates (SG), and three distinct non-affected areas were annotated for transcript analysis (Supplementary Fig. 1D). These annotated areas displaying similar morphological features

**Fig. 4** Identification of transcripts co-localizing with *M. tuberculosis* in tissue sections. **a** The networks of co-expressed transcripts in one representative lymphoid area of granulomas at 8 and 12 wpi are shown. **b** The networks of co-expressed transcripts in one representative epithelioid area of granulomas at 8 and 12 wpi are shown. **c** The auramine–rhodamine-stained *M. tuberculosis* bacteria were aligned with in situ transcript signals in tissue sections at 8 wpi. *M. tuberculosis* bacteria were identified by an automated cellprofiler pipeline and a 30 μm radius around those identified is depicted. **d** The *Inos* and *Cd68* transcript signals at <30 μm from identified *M. tuberculosis* bacteria are shown together with *M. tuberculosis* and DAPI staining for the same selected region as in **c**. Scale bar: 200 μm. **e** The sequences located within a 3, 10, 30, 100, and 200 μm radius from *M. tuberculosis* bacteria were identified. The frequency of each sequence within a given distance was determined in relation to the total transcript count for that distance. The fold increase of this frequency with respect to that observed for the total lung section is depicted. Thus, whether a certain transcript is over- or under-represented within the defined distances from *M. tuberculosis* was determined. **f** The *Cd68* and *Inos* transcripts located at <30 μm are shown aligned with the H&E staining of one representative section (of three sections). Note that most of the extracted sequences are present in the epithelioid region in relative proximity to the lymphoid areas. Scale bar: 1000 μm. Source data are provided as a Source Data file

(from the same annotated "group") showed related transcript profiles and were distinct to those from other groups (Fig. 7d, e). The center of the encapsulated granulomas (GC) contained a lower *Actb* mRNA density as compared with other areas (GO, NOG, or SG) reflecting the necrotic extension (Fig. 7d).

GC contained higher densities of *Foxp3* and *Il10* mRNA, expressed by regulatory T cells (T_regs), as compared with the rim or with other regions (Fig. 7c–f). GC also showed higher levels of *tcrg* (coding for the γ-chain of the T-cell receptor of γδ T cells), *Cd3e*, and *Il6* transcripts (Fig. 7d, e). Instead, GO showed higher frequencies of *Tcrb* (coding for the β-chain of the T-cell receptor of αβ T cells), *Cxcr3*, *Cd8*, *Cd68*, and *Inos* mRNA than GC (Fig. 7d–f).

Levels of *Cd68*, *Inos*, *Cxcr3*, *Tcrb*, and *Ifng* transcripts were higher in NOG than in the center (GC) or the rim (GO) of the large and the smaller encapsulated granulomas (SG) (Fig. 7c–g). *Cd19* mRNA clusters were not observed in the lung of *M. tuberculosis*-infected C3HeB/FeJ mice (Fig. 7c).

An unbiased hexbin analysis also showed a clear distinction of clusters in their transcript contents that aligned in the necrotic cores (red) and the surrounding areas of the necrotic and the cellular granuloma (green cluster, Fig. 7h). The centroid expression of *Foxp3*, *Il6*, and *Il10* transcripts were highest in the red cluster in agreement with the analysis of annotated areas (Fig. 7i).

A network analysis showed co-expression *Foxp3*, *Tcrg*, and *Il6* mRNA in the necrotic granuloma (Supplementary Fig. 6E). Instead, NOG in C3HeB/FeJ mice showed *Cd68-Inos* interactions resembling interactions described for the epithelioid regions of C57BL/6 granulomas (Supplementary Fig. 6F and Fig. 4b).

Altogether, encapsulated granulomas in C3HeB/FeJ mice contained a center enriched for transcripts associated with immune suppression, while cellular granulomas contained networks associated to protective responses.

## Discussion

Studies using TB granulomas in man came to an end in the 1950s with the introduction of antibiotics and the decline of lung lobectomies[16]. Research interest relocated from morphological histopathology to immunology, molecular microbiology, and genetics studied in isolated cells and animal models. Although the understanding of the biology of infection in these areas has increased immensely, that of the granuloma structure has advanced at a slower pace. Although the novel methods have recently improved the molecular and cellular dissection of the granuloma, this has largely been done in homogenates or isolated cell suspensions in which the histological context is lost. Here, the localization of 34 immune transcripts in lungs during infection with *M. tuberculosis* was resolved at cellular resolution and aligned with the tissue topology generating high-resolution images of the immunological landscape of granulomas.

The in situ sequencing method displayed a high signal-to-noise ratio of fluorescent signals and high specificity of analyzed sequences allowing target recognition at the single nucleotide level, as previously reported[9]. One limitation of our study is the small number of independent sections examined. This is due to the extensive image acquisition and data processing. Statistics were performed by comparing frequencies of transcripts in different areas of the lung or in different regions of the granuloma but also by comparison of determinations in consecutive sections, which confirmed the specificity of the signals. The results presented were confirmed in one independent sample for each condition.

Histopathological analyses have determined that the formation of murine and human granulomas is regulated by the orderly recruitment of immune cells[17]. Here we show that most of the immune transcripts identified located within the granulomas. Whereas the frequency of transcripts expressed by innate immune cells (i.e., *Tnf*, *Il6*, *Inos*, *Il12*) in the granuloma at different time points after infection was similar, several transcripts expressed by T- or B cells (*Cd3e*, *Cd19*, *Ccr6*, *Ifng*) showed an increased relative localization in the granuloma at later time points. Networks of co-expressed transcripts showed an increased complexity of interactions in the granulomas at 8 and 12 compared with 3 wpi. In line with this, the initiation of *M. tuberculosis*-specific T-cell responses that limit bacterial growth has been shown to occur later than immune responses stimulated by other infections[18].

An unsupervised analysis using clustering estimation of transcript expression patterns in the infected lung represented in hexbin maps showed differences in transcript levels (3 and 8 wpi) early, but a distinct relative abundance of specific transcripts in the different clusters was observed only late after infection. At 12 wpi, an over-representation of *Cd19* mRNA was noted in one of the clusters. *Cd19* mRNA localized in lymphoid cell areas that were detected at 8 and 12, but not 3 weeks after infection. In such lymphoid areas, *Cd19* and T-cell transcripts were co-expressed together with *Ccr6* mRNA in most granulomas, indicating close interactions between T cells and B cells in the lymphoid areas. Notably, CCR6 is expressed on interleukin (IL)-17-expressing cells such as T_H17 cells and ILC3s, and participates in their homing to mucosal microenvironments[19,20], and RORγt is the key transcription factor required for T_H17 differentiation and IL-17 expression[21]. CCR6 might also be transiently expressed in germinal center B cells and dendritic cells[22,23], and has been shown to be essential in the development of lymphoid follicles[24]. Both T_H17 cells and B cells are important for the formation of tertiary lymphoid organs[25,26], including lymphoid aggregates in granulomas during infection with *M. tuberculosis*[27]. T_H1-associated molecules such as *Cxcr3* and *Ifng* were also co-expressed in networks of the lymphoid-rich areas.

Ectopic lymphoid tissues or tertiary lymphoid organs are present in tissues during several infections, autoimmune diseases,

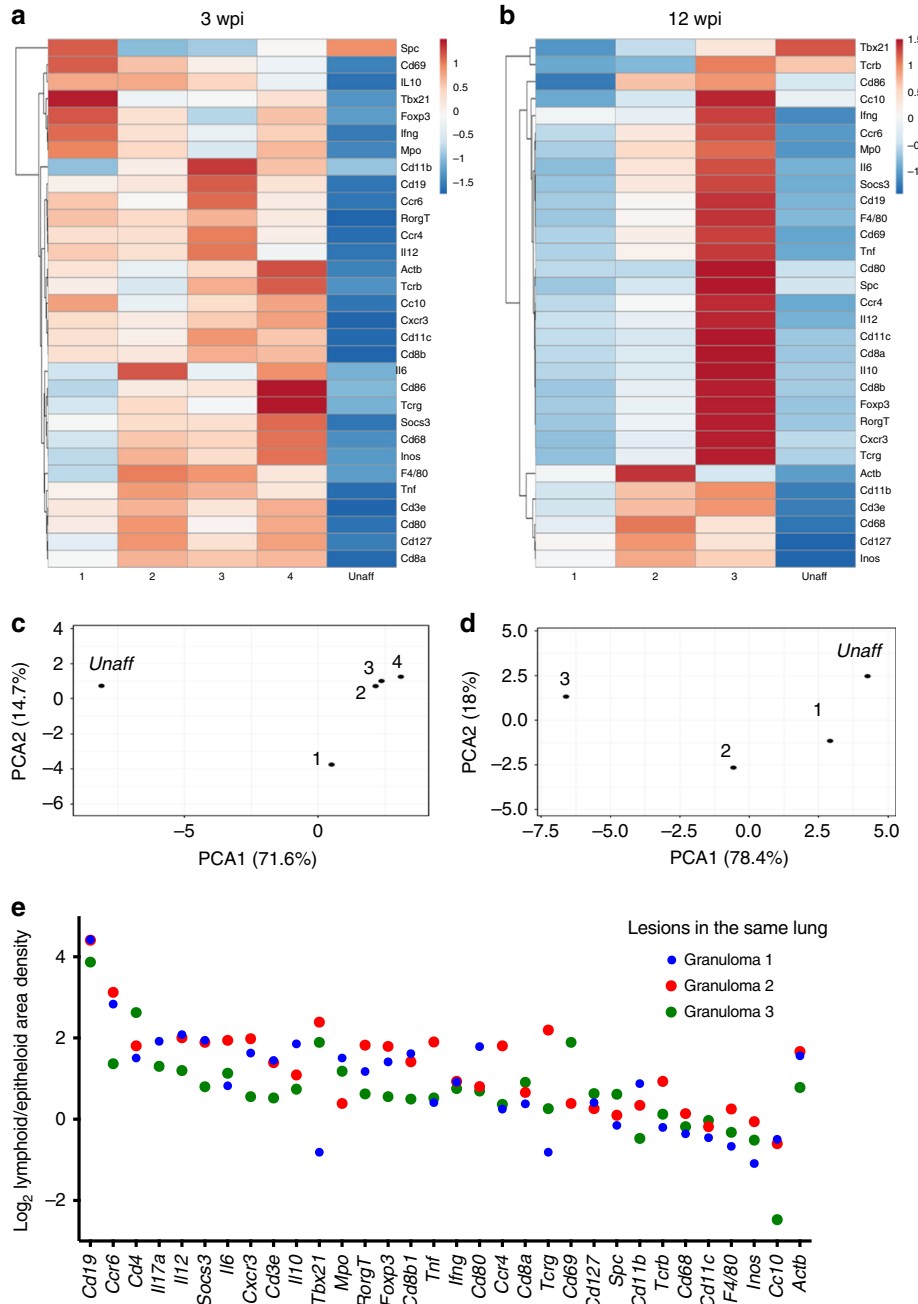

**Fig. 5** Heterogeneity of granulomas in the same lung. **a–d** Heat map analysis depicting the transcript density in different granulomas as defined in Supplementary Fig. 1 at 3 (**a**) and 12 (**b**) wpi normalized per area. To simplify the description of the data, results were processed by a multivariate principal component analysis (**c, d**), showing that although some granulomas cluster together with respect to their transcripts titers, others segregate from these as well as from the unaffected area. One representative result of three consecutive sections per time point is shown. **e** The density of sequences in lymphoid and epithelioid areas of individual granulomas of one tissue section 12 wpi was extracted. The ratio of sequence density of epithelioid and lymphoid areas per granuloma in the three different granulomas in the same lung was quantified and depicted. Source data are provided as a Source Data file

and tumors[28]. Different to ectopic lymphoid tissues in other pathologies[29], a clear distinction of T-cell areas and B follicles is not evident in lungs from *M. tuberculosis*-infected C57BL/6 mice. Lymphocytic aggregates in TB granulomas containing proliferating B cells, CXCR5+ T cells, follicular dendritic cell networks, and high endothelial venules[14] have been described in mice[30], non-human primates[31], and humans[13]. Chemokine networks involved in developing lymphoid structures have been associated with protection in experimental infections with influenza and *M. tuberculosis*[14,30]. The presence of these cytokines and the observed close contact between B cells and T cells suggest that

either specific antigens might be presented to primed T cells, or that naive T cells may be primed in these areas. In several lymphoid areas the expression of *Il12* mRNA was observed. IL-12p40 is a constitutive member of IL-12 and IL-23, cytokines secreted by antigen-presenting cells and required for T_H1 and T_H17 differentiation, respectively, which takes place in the lymph nodes. In line with this, antigen-specific T cells were generated during *M. tuberculosis* infection in lungs from mice devoid of lymph nodes due to the deletion of lymphotoxin-α[32]. Together, the lymphoid-rich areas may constitute a structure where host immune responses to *M. tuberculosis* are orchestrated.

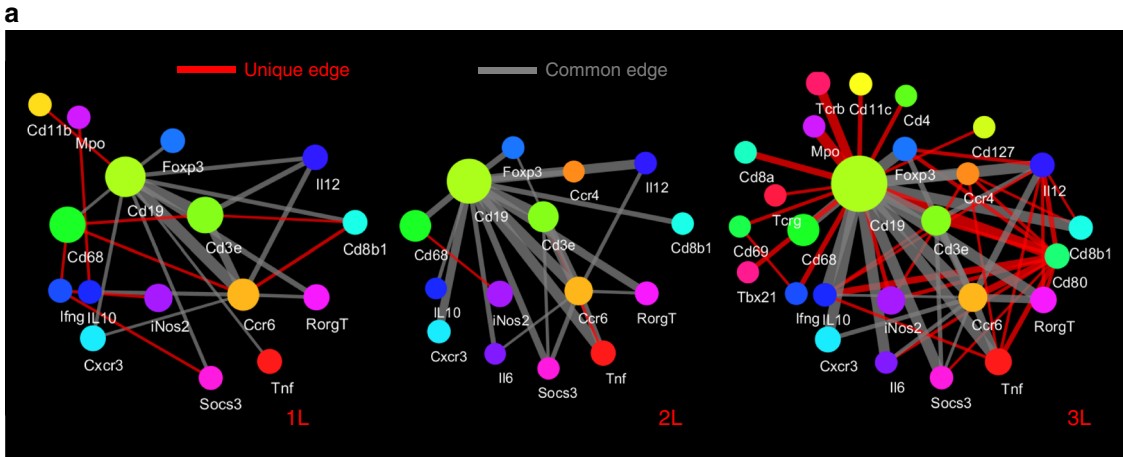

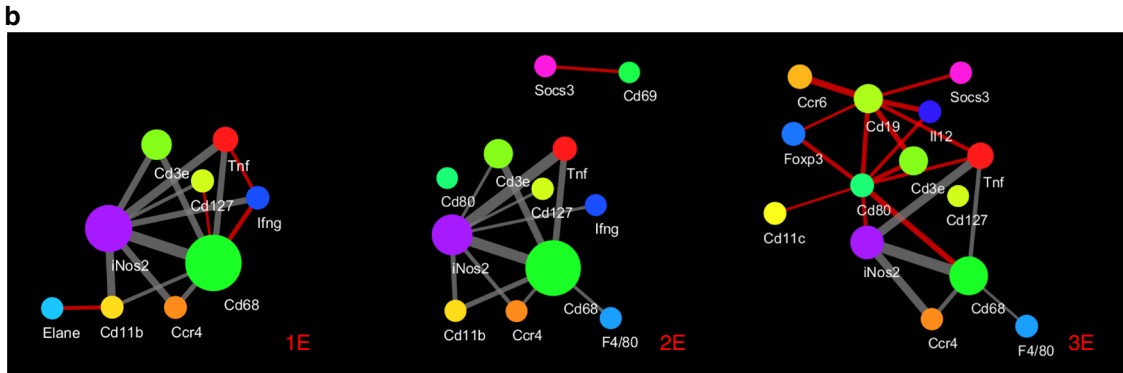

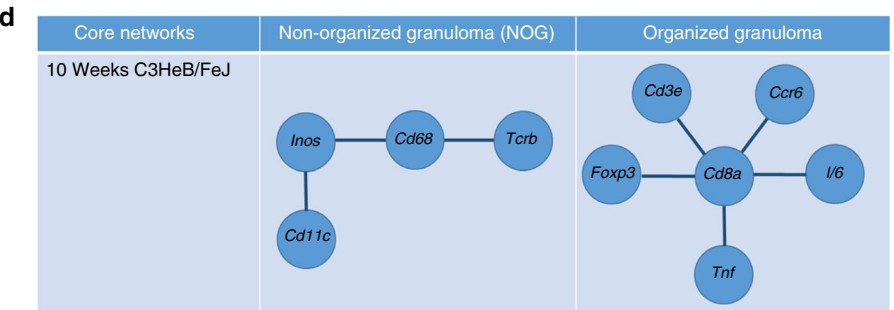

T cells also located in the epithelioid areas of the granuloma. In these areas, the main clusters included *Cd68*, *Tnf*, and *Inos* but also *Cxcr3* and *Ccr4* mRNA expressed by T_H1 and T_H2 cells, respectively[33]. This suggests both T_H1 and T_H2 cells penetrate into the epithelioid compartment. In agreement with the observed compartmentalization, CD68+ macrophages in the human tuberculous tissues do not form aggregates with B lymphoid clusters[17].

Our data indicate the preferential presence of *Inos*, *Cd68*, *Tnf*, and *Cd11b* mRNA sequences at subcellular distances (<3 μm) to

**Fig. 6** Heterogeneity of transcript networks in areas of the granuloma during infection with *M. tuberculosis*. **a**, **b** Diversity of interacting networks of transcripts in the lymphoid (**a**) and epithelioid (**b**) area of three granulomas from one representative lung 12 wpi is depicted. Nodes in the network represent unique transcripts and node size is proportional to the number of transcript detections. Edges represent significant spatial co-expression between transcripts. The more statistically significant the co-expression is, the greater the weight (thickness) of the edge in the network. Connections that occur in at least two of the three networks per region appear in gray, unique connections in red. **c**, **d** Core networks, defined as co-expressed transcripts at <10 μm in consecutive sections found in all annotated regions (Supplementary Fig. 1) for each time point, calculated as described in the Methods section, are depicted. Diverse core networks were defined for whole granulomas, for epithelioid and lymphoid regions of the granulomas at different time points after *M. tuberculosis* infection in C57BL/6 mice (**c**). The core networks from non-organized and organized granulomas C3HeB/FeJ mice are also shown (**d**). No core networks for unaffected regions could be found

*M. tuberculosis* and strongly advocate that the lesion contains infected activated macrophages. These transcripts localized in the epithelioid areas of the C57BL/6 mice granuloma, usually in proximity to the lymphoid regions containing cellular populations that might perpetuate the activation of these inflammatory macrophages. iNOS-derived NO plays a key role in host defense against mycobacteria[34]. A variety of *M. tuberculosis* determinants allow the evasion from macrophage bactericidal mechanisms[35]. Relevant for our data, *M. tuberculosis* can detect oxidative-nitrosative-induced changes in the host environment[36] and respond by producing proteins that limit the toxic effects of these changes, including enzymes that repair NO-damaged bacterial proteins at the proteasome level[37–39]. *M. tuberculosis* also inhibits iNOS recruitment to the phagosome to ensure its intracellular survival[40]. This might explain the mycobacterial localization in TNF- and iNOS-expressing macrophages. Transcripts expressed by airway or bronchiolar epithelial cells were negatively associated with *M. tuberculosis*, despite the interaction of the bacteria with airway and alveolar epithelial cells has been shown[41–43].

Structural, microbiological, and immunological heterogeneity of TB granulomas in the same lung has been described in animals and humans[44–46]. Both heat maps and the co-expression network revealed significant variances in transcript localization. We also show a significant variance between different lymphoid and epithelioid areas of different granulomas, indicating that the differences are not only due to a diverse ratio between both areas. Thus, heterogeneity of transcript networks between histologically similar granulomas may represent a limited microenvironment that might be influenced independently from the others in the same lung by the quality of the local immune response and the level of inflammation. However, common core networks that are unique for each region, time point, and mouse strain were here defined.

C3HeB/FeJ mice developed encapsulated and non-encapsulated granulomas after *M. tuberculosis* infection, also displaying distinct transcript patterns. The encapsulated granuloma contained a hypoxic necrotic central area surrounded by a thick fibrotic capsule separating the lesion from other lung areas[15,47]. The increased levels of *Foxp3* and *Il10* mRNA found in the encapsulated granuloma center support a local suppression of pulmonary inflammatory cells. The presence and suppressive function of *Foxp3* expressing $T_{regs}$ in granulomatous responses has been previously shown[48–50] and we suggest also that the localization of these cells coincides with regions of high bacterial density. Similar to our findings, granulomas from children with tuberculous lymphadenitis showed increased numbers of CD4$^+$FoxP3$^+$ T cells, whereas CD8$^+$ T cells surrounded the granuloma[51].

IL-10 is produced at the site of active TB in humans and mice, and in both an inhibitory role of protective immunity by IL-10 has been suggested[52–55]. $T_{regs}$, interferon-γ-secreting CD4+ cells, neutrophils, and macrophages have all been shown to produce IL-10 during mouse or human TB[53,56,57]. In association with lower bacterial levels, NOGs showed a higher frequency of *Tcrb*, *Ifng*, *Inos*, *Cd68*, and *Cd11b* mRNA, which code for proteins of

activated macrophages and $T_H1$ cells, as compared with the necrotic granuloma.

A combination of laser microdissection and mass spectrometry showing the abundance of proteins in different compartments of granulomas from lungs resected from TB patients and from rabbits has described physically segregated signals within each granuloma[58]. This and/or other spatially resolved proteomic and transcriptomic approaches[59,60] such as in situ sequencing could complement each other in delineating the molecular map of the TB granuloma.

In summary, granulomas showed increasing complexity of co-expressing molecular networks with time after infection. Such increased complexity was due to the presence of adaptive immune transcripts, some of which co-expressed in lymphoid clusters. *M. tuberculosis* colocalized with transcripts from activated macrophages in NOGs. Encapsulated granulomas showed necrotic centers with transcripts associated with anti-inflammatory responses. Instead, those in the outer rim of necrotic granulomas or in cellular non-organized lesions in the same lung showed higher abundance of pro-inflammatory networks. Morphologically similar lesions showed highly diverse transcript networks with common cores.

The use of in situ sequencing in paraformaldehyde-fixed samples eliminates the risk of handling live TB tissues and enables the study of human TB granulomas from either autopsy libraries or surgical biopsies from all stages of disease, including sections from TB patients with co-morbidities such as diabetes, obesity, or in immunosuppressed individuals to HIV, cancer treatment or aging. This will allow understanding molecular differences in the buildup of the granuloma in relation to the clinical status of the patients, providing a high-resolution picture of the TB granuloma in different immunological settings.

## Methods

**Mice**. C57BL/6 mice were purchased from Janvier labs; C3HeB/FeJ mice were received from Igor Kramnik (BU, Boston, MA). All mice were housed and handled at the Astrid Fagreus Laboratory, Karolinska Institute, Stockholm, under specific pathogen-free conditions and according to directives and guidelines of the Swedish Board of Agriculture, the Swedish Animal Protection Agency, and the Karolinska Institutet.

**Infection and infectivity assay**. Mice were infected with ~200 *M. tuberculosis* Harlingen strain by aerosol using a nose-only exposure unit (In-tox Products). At the indicated time after infection, mice were killed and lungs extracted and fixed in 4% buffered paraformaldehyde for 24 h. Fixed left lungs of mice experimentally inoculated with *M. tuberculosis* were paraffin-embedded. From each lung sample, ten longitudinal 8 μm sections (along the long axis of the lobe) were obtained and stored at −80 °C. Slides were paraffin-removed and dehydrated directly before further processing. Sections were stained with H&E after the in situ sequencing procedure. Sections from C57BL/6 mice 8 wpi were also stained with Auramine–Rhodamine T staining mycobacterial lipids following the instructions of the manufacturer (BD).

**Immunohistochemistry**. For immunofluorescent staining, deparaffinized, rehydrated, and demasked sections were blocked in 5% bovine serum albumin (BSA) for 30 min, washed, and incubated with 1/50 rat anti-mouse B220 antibody (clone RA3-6B2, BD Pharmingen, Catalog Number 550286), 1/10 rat anti-mouse CD3

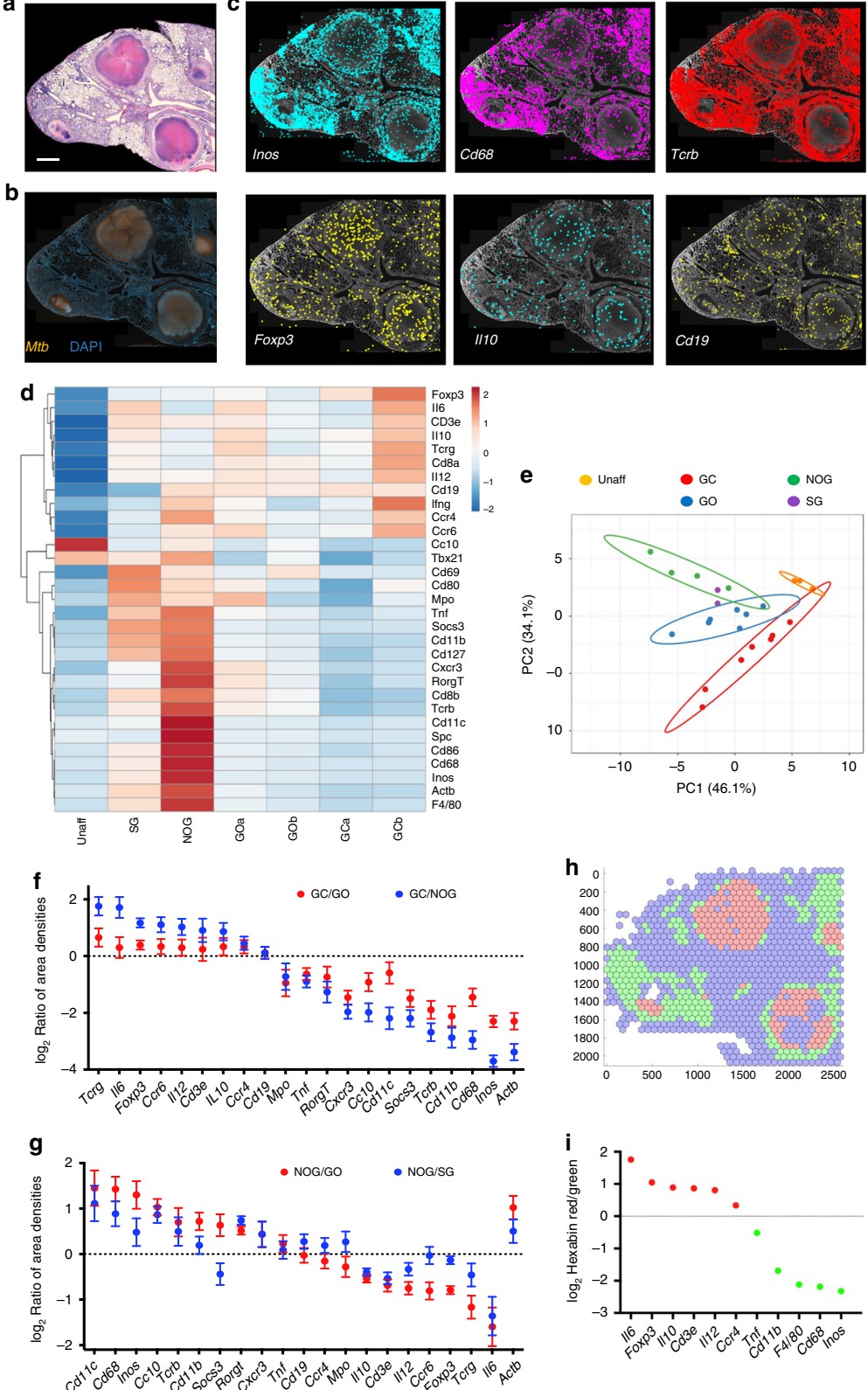

(clone CD3- 12, Abcam, ab11089), and 1/50 rabbit anti-mouse polyclonal CD68 (Abcam, ab125212) in 1% BSA, 0.3% Triton-X phosphate-buffered saline (PBS) for overnight at 4 °C. After washing in PBS, sections were incubated for 1 h at room temperature with 1:100 dilution of secondary rhodamine red-labeled donkey anti-rabbit IgG and Alexa Fluor 488-labeled mouse anti-rat IgG (both Jackson ImmunoResearch, Catalog Number 711-295-151, 212-545-168) in antibody buffer. Slides were then washed and mounted in Vectashield with DAPI.

**In situ sequencing technique**. The in situ sequencing technique was performed as previously described[9]. Briefly, 8 μm tissue sections from paraformaldehyde-fixed and paraffin-embedded lungs were obtained, mounted on microscope slides (Superfrost Plus) and stored at −80 °C until further processing. After partial digestion with pepsin at 37 °C for 30 min, tissue sections were washed in PBS, then dehydrated by passing them through 70% and 100% ethanol and air-dried. Gaskets (SecureSeal Hybridization chambers, Grace Bio-Labs) were glued onto the slides

**Fig. 7** In situ sequencing in encapsulated granulomas. **a**, **b**. H&E (**a**) and Auramine–Rhodamine stain of *M. tuberculosis* bacteria together with DAPI staining (**b**) of a C3HeB/FeJ lung section 10 wpi. Scale bar: 1000 μm. **c** In situ signals of *Inos*, *Cd68*, *Tcrb*, *Foxp3*, *Il10*, and *Cd19* mRNA transcripts in a lung section from *M. tuberculosis*-infected C3HeB/FeJ, aligned with the DAPI staining, are depicted. **d**, **e** Heat map analysis (**d**) depicting the sequence density in the annotated areas of region 1 of the C3HeB/FeJ lung (Supplementary Fig. 1D). A multivariate principal component analysis of signals shows proximity between those in annotated areas from all regions sharing histopathological features, but the distances between those of different kinds (**e**). The predictive ellipses displayed have a 95% probability that a new observation from the same group will fall inside the ellipse. The areas correspond to the center (GC) and edge (GO) of encapsulated granulomas, non-encapsulated granulomas with relatively low mycobacterial density (NOG), small-encapsulated granulomas with high bacterial numbers (SG), and an unaffected area (Unaff). **f** The density of sequences in the different areas of the C3HeB/FeJ lung were extracted. The $\log_2$ ratio of sequence density ± SEM in the center (GC) and edge (GO) of the encapsulated granuloma and in the center vs. the non-organized granuloma (NOG) are depicted. The correlation between the GC/GO and GC/NOG is significant (Pearson's $p < 0.001$; $r$: 0.67). **g** The $\log_2$ ratio of sequence density ± SEM in the non-organized (NOG) vs. the surrounding area from the encapsulated granuloma (GO) or the small necrotic granuloma (SG) are depicted. The correlation between the NOG/GO and NOG/SG is significant (Pearson's $p < 0.002$; $r$: 0.65). **h** The sequences in lungs from C3HeB/FeJ in 70 μm radius hexbins were clustered into three different expression patterns. These overlap with regions with encapsulated and non-encapsulated granulomas or with unaffected lung areas (Supplementary Fig. 1D). **i** The mean centroid normalized sequence counts in each hexagon was compared for the clusters. The ratio of sequence densities in red and green clusters is depicted. Source data are provided as a Source Data file

such that they form a sealable reaction chamber enclosing the tissue. The mRNA in such sections was in situ reverse-transcribed to complementary DNA using random decamer primers and primers that partially overlap with the recognition sequence of the padlock probes (Supplementary Table 2). After reverse transcription, an additional crosslinking step was performed (4% paraformaldehyde at room temperature for 45 min), followed by degradation of the mRNA strand and hybridization of padlock probes to the remaining cDNA strand. A ligase in the reaction mix catalyzes circularization of hybridized padlock probes. Multiple padlock probes were designed for each of the 34 genes of interest such that they detect non-overlapping, transcript-specific sequences (Supplementary Table 3). Every set of padlock probes for a given transcript carries a unique four-base barcode, which is used for identification[9].

In situ sequencing substrates are generated in a RCA reaction, which is primed by the cDNA using circularized padlock probes as template. Resulting RCA products (RCPs) are subjected to sequencing by ligation. An AlexaFluor750-conjugated sequencing anchor oligo is hybridized immediately adjacent to the barcode sequences of all RCPs (Supplementary Table 4). Four nonamer libraries are added to the reaction, each containing random nucleotides and one specific base (A, T, G, or C) at a fixed position. The libraries are designed such that every base corresponds to a specifc dye. In an enzymatic DNA ligation reaction, the anchor primer is joined to species of the nonamer pool that best match the barcode. After performing the ligation and imaging, the RCPs are reset by stripping off anchor primer and nonamers, and a new cycle is initiated. The process is repeated until all four positions of the barcodes have been interrogated. Imaging was performed in an Axio Imager Z2 epifluorescence microscope (Zeiss) by acquiring Z-stacks of overlapping tiles that together cover the tissue section (10% overlap). Image stacks were merged to maximum-intensity projections (Zen software).

**Image analysis**. A fully automated image analysis pipeline was performed using CellProfiler (v.2.1.1) calling ImageJ plugins for image registration (all scripts in repository under https://doi.org/10.6084/m9.figshare.7663010, Cellprofiler pipelines: Blob identification)[9]. Briefly, images from all four sequencing cycles were aligned utilizing their general stain for RCPs saving *x* and *y* coordinates, as well as fluorescence intensities for each RCP in their four base positions to a .csv file and decoded using a Matlab script (Matlab scripts: InsituSequencing). For each RCP and hybridization step, the RCP was assigned the base with the highest intensity. A quality score was extracted from each base, defined as the maximum signal (i.e., intensity of assigned letter) divided by the sum of all signals (letters) for that base that ranges from 0.25 to 1. A value close to 0.25 means poor quality (similar signal for all letters), whereas a value close to 1 means that the signal of the assigned letter is strong above a low background. A fixed threshold of 0.45 that gave around 3% unexpected reads was applied and positional data were saved in a .csv file (repository, Lung csv files).

For signal visualization, the InsituSequencing Matlab script was used to plot selected transcripts on H&E/DAPI-stained images of the analyzed section and to calculate two log transformations of kernel density estimations for each mRNA species (Matlab Scripts: Density Estimation). The number of transcript signals were extracted from the whole tissue and from manually selected regions based on pathological features (Supplementary Fig. 1) for further analysis.

For an unsupervised analysis of our spatial data, we used a Matlab script (HexbinClustering) that generated *k*-mean clusters for a given number of clusters and size of hexbins. This iterative learning algorithm aims to find groups in the data by clustering them based on similarities in their transcript expression levels. Transcript counts in every hexbin were normalized by their maximum counts. For each RNA species a cluster centroid (=mean normalized expression level) was computed. The minimum number of clusters that rendered differential results were analysed.

**Heat map and principal component analysis**. Extracted transcript reads were normalized to the area and uploaded to ClustVis, a web tool for visualizing clustering of multivariate data using heat maps and principal component analysis (https://biit.cs.ut.ee/clustvis/)[61]. The heat map used linear expression data and is row-centered, and unit variance scaling (the SD was used as the scaling factor) was performed for rows. ClustVis calculates principal components using one of the methods in the pcaMethods R package and plots heatmaps using heat map R package (version 0.7.7).

**Colocalization analysis**. We used two applications in Cytoscape, an open source software platform (http://apps.cytoscape.org). InsituNet, a cytoscape app that converts in situ sequencing data (above described .csv file with transcript name, x and y coordinates) into interactive network-based visualizations, where each unique transcript is a node in the network and edges represent the spatial co-expression relationships between transcripts, was applied to identify co-expressed transcripts[12]. Co-expressed transcripts were defined in a range of <10 μm and their statistical significance was assessed by label permutation and corrected for multiple testing by the Bonferroni method. Those networks were imported into DynetAnalyzer to further compare networks and extract core networks[62].

**Identification of bacteria and surrounding transcripts**. After in situ sequencing, sections were stained for bacteria with Auramine–Rhodamine T and also hybridized with the fluorescence-labeled anchor primer for the general stain. As for individual base positions, multiple focal plane images were acquired and merged to maximum-intensity projections. Bacteria images were aligned to the corresponding four base position images customizing the Cellprofiler pipeline from above (https://doi.org/10.6084/m9.figshare.7663010, Cellprofiler pipelines: Bacteria identification). In short, bacteria were identified after subtraction of autofluorescence based on global Otsu thresholding; intensities of RCPs were extracted as before with the additional information whether they were located within the assigned distance (3, 10, 30, 300, and 600 μm) to identified bacteria. Matlab scripts were used for decoding and plotting of transcript signals (https://doi.org/10.6084/m9.figshare.7663010, Bacteria csv files).

**Statistical analysis**. Co-expressed transcripts in networks were defined in a range of <10 μm and their statistical significance was assessed by label permutation and corrected for multiple testing by the Bonferroni method in the InsituNet app.

The densities of transcripts in granulomas or unaffected regions were analyzed by $p < 0.01$ $\chi^2$-test using Prism Graphpad.

Differences in relative transcript density in the lesions at different time points after infection were analysed by unpaired Student's *t*-test Prism GraphPad.

The ratio of area densities of different annotated areas in the section in Fig. 7f, g were significantly correlated (Pearson's test).

ClustVis was used to generate prediction ellipses that show the area with a 95% probability that a new sample belonging to the same group will be contained in it, providing a "graphical multivariate analysis of variance test".

**Reporting summary**. Further information on research design is available in the Nature Research Reporting Summary linked to this article.

## Data availability

Raw images for 12 weeks after infection, positional csv files, DAPI images for plotting, and high-resolution H&E staining of in situ-sequenced lungs for all time points can be found under https://doi.org/10.6084/m9.figshare.7663010. Additional raw images will be shared after request. All scripts for the in situ sequencing processing pipeline and downstream analysis can be found under the same DOI. The source data underlying Figs. 1B, 2C, 3D, 4E, 5A–E, 7D–I and Supplementary Figs. 2E and 5C are provided as a Source Data file.

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

## Acknowledgements

We thank the expert help of the staff of the Astrid Fagreus animal house for this study and the comments from Dolores Gavier-Widén. We acknowledge the "In situ sequencing" infrastructure facility unit at Science for Life Laboratory, Solna, Sweden, for technical assistance with the in situ sequencing experiments. This study was supported by the Swedish Heart and Lung foundation 2015-17/20140641, the Swedish Research Council 2015-02296, the Swedish Institute for Internationalization of Research (STINT) 4-1796/2014 to MER, the European Community H2020 (grant number 643558), and the Karolinska Institutet.

## Author contributions

Investigation: B.C., T. H. and X.Q. Writing original draft: M.E.R. Review and editing: B.C., M.N., M.E.R., T.H. and I.K. Conceptualization: B.C. and M.E.R. Resources: M.N. and I.K. Funding acquisition: M.N. and M.E.R.

## Additional information

**Competing interests:** The authors declare no competing interests.

