## [Peer Review File · Nature Communications]

Reviewers' comments:

Reviewer #1 (Remarks to the Author):

Carow et al. report on temporal localization of immune transcripts in the tuberculosis granuloma. They performed simultaneous in situ sequencing of 34 immune transcripts in lung sections from Mycobacterium tuberculosis-infected and C57BL/6 C3HeB/FeJ mice, the latter containing both encapsulated and non encapsulated granulomas. The RNAs analyzed are selected to represent various cell types expected in the granulomas. The rolling circle in situ amplification appears to be sufficiently robust to align specific RNA transcripts with cell type and subcellular proximity to M.tb. With extensive imaging, this study provides a detailed view of the cellular structure of granulomas, inferred from transcript profiles, providing new information about the time course of granuloma maturation, transcript compartmentalization, and heterogeneity. For example, the RNA expression landscape shows that encapsulated granulomas have necrotic centers with immunosuppressive transcripts and those in rims associated with activated T cells and macrophages. Likely representing activated infected macrophages, cd68, cd11b, tnf and socs3 mRNAs were enriched in close proximity to M.tb. On the other hand, ccr6, cd19, il12, cd3, cxcr3 and cd8b mRNA are enriched in lymphoid over epithelioid areas. These are all interesting observations that provide new insights, with good discussion how the data fit into the abundant literature on granuloma structure and function.

In the discussion, the authors cite a study using a combination of laser microdissection and mass spectrometry to analyze the abundance of proteins in granulomas regions – maybe this study could have been better represented in the Introduction, to highlight what new information is expected from the RNA profiles.

While the study is largely descriptive, the authors propose that “each granuloma probably represents a limited microenvironment that might be influenced independently from the others .. by the quality of the local immune response and the level of inflammation”, “Our findings emphasize the importance of analyses at the tissue-level rather than in blood cell suspensions or in tissue homogenates”, and “the lymphoid-rich areas may constitute the structure where host immune responses to M. tuberculosis are orchestrated. This has implications for the design of mucosal vaccines.” These are rather sweeping statements that lack some granular understanding of what exactly is new, and what are the implications for vaccine design?

Wolfgang Sadec

Reviewer #2 (Remarks to the Author):

Work by Carow, Hauling et al utilized in situ sequencing of 34 transcripts to define spatio-temporal immune landscapes in lung sections from Mtb-infected mice. They show that transcript clusters do not differ much by density earlier on during the infection, but that there is more compartmentalization of transcript densities within granulomas at later time points after infection. Using encapsulated mice granulomas, the authors also show that granuloma centers are enriched with transcripts associated with immune suppression whereas cellular granulomas are enriched with transcripts associated with protective responses. The work is novel in the use of a highly sensitive in situ imaging of mRNA species to characterize time-specific immune diversity in TB granulomas. Their findings on spatial segregation of various immune factors is complementary to and extends results on human and non human primate TB granuloma heterogeneity/variability recently published using PET imaging, flow cytometry transcriptomics and proteomics. The impressive

technology applied here is a major strength of the manuscript, and it will definitely be of great interest in the community and to the wider field.

All in all, the manuscript is well written and presented.

Below are the issues that can be addressed to improve the manuscript:

Major comments:

- What is the basis on which the 34 transcripts were chosen for analysis in this study? Is there a rationale why this specific list was considered?
- Throughout the manuscript, the authors show image analysis of one lung tissue sample at each time point post infection (3, 8, 12 weeks). Are these a representation of more lung samples analyzed at these time points? If so, are the image analyses reproducible in the other lung samples? This
- It would be interesting to evaluate the performance of the in situ sequencing in heterogeneous lesion types (rather than just one lung section in fig 7) produced in the C3HeB/FeJ mice. Considering that previous work has shown the presence of multiple lesion types with vastly different microenvironments in C3HeB/FeJ mice following aerosol infection with Mtb (Ervin et al, *Disease Models & Mechanisms* (2015) 8, 591-602).
- In figure 3C, the authors show that cd3 mRNA co-localizes with cd68 mRNA at the epithelioid areas. This localization observation should be strengthened by immunohistochemical validation using antibodies/markers for epithelioid cells. This could be done the same way as in figure 3b, where the authors beautifully confirmed specificity of the in situ sequencing (for cd19 mRNA) by using immunohistochemistry staining of B cells.
- More specific details are required to allow better understanding of the methods as well as the ability for a researcher to reproduce the work. For example; what is the number of the lung tissue samples analyzed for each time point, how many mice were used in the experimentation (at each time point), what Mtb strain was used for the aerosol infection?
- It would be interesting to evaluate if any of the transcripts compartmentalized at later time points would influence disease outcome. The authors should discuss more on potential clinical implications of these findings in the context of TB disease control/progression.
- Also, considering differences between mouse and human TB granulomas, the authors should discuss if the temporal and spatial transcripts observed in this study can be extrapolated for translational interventions in TB patients.

Minor comments:

- Please put scales of the lung images utilized throughout the manuscript.
- Numbering on the legends for Figure S2 needs to be fixed.
- Under 'maturation of the granuloma', authors state that 'all but 4 mRNA species (cd4, cd40l, elane and il17a) showed increased localization'. I can't locate CD40L and elane from the density plots in figure 1 and supplemental figure S2e.
- Under 'maturation of the granuloma', authors list several transcripts expressed by myeloid cells at a similar frequency, and then increased enrichment for transcripts that are associated with

adaptive immune responses. Among the list of transcripts associated with adaptive immune responses, the authors include CD68. This molecule is actually a myeloid-specific marker and more abundant in macrophages and monocytes, and not adaptive immune response (<https://www.ncbi.nlm.nih.gov/pubmed/?term=CD68+Chistiakov+DA>).

- Figure 5b: According the heatmap clustering tree at 12 wpi, 'U' and '3' appear to be clustered together; while the actual expression intensities are opposite (3 is entirely red and U is entirely blue). This needs to be reconciled?
- Figure 7 legend seems incomplete

Reviewer #3 (Remarks to the Author):

In this manuscript, Carow et al aim to resolve granuloma spatial structure and uncover granuloma heterogeneity in tissues by combining histology staining and in situ sequencing - a technology able to image RNA transcript localizations on fixed sections. Here 34 transcripts have been simultaneously monitored and a tailored bioinformatics tool (coined 'insituNet') has been used to decipher the transcript co-localization network. After infection with a Mycobacterium tuberculosis, the immune system react by forming multi-cellular structures called granuloma that aims to contain the bacteria. These structures are highly diverse and in a same section, granuloma can be highly different and the method introduced here by the authors is one of the most advance to perform spatial transcriptomics. The application of in situ sequencing to granuloma structure is highly relevant and timely to answer the question of heterogeneity in host-pathogen interaction.

The paper is well written and the analysis is conducted in a clear sequential manner. The analysis is greatly done. Although impressive from a methodological point of view, I feel that the biological outcome of the study is not clearly explained and need to be reinforced. The way that data are displayed can clearly be improved.

- The rational of the selection of the 34 tracked transcripts could be better explained. For example: what genes refer to "macrophage activation"?
- In order to quantify the specific transcripts in the granuloma region the authors several metrics: "fold increase over unaffected area" (s-fig2), Density plots (fig1A), "log2 mean fold Granuloma/unaffected area" (fig1B), ': this is overall very confusing. Overall page 6 line 2-14: this part is cryptic. Why Fig 1B does not contain the '8 weeks' time point? Other example : is really cd68 above actb level? Actb could be added in Fig.1B.
- Transcript co-expression is not clear for me. Why actb is not in? Just by abundance the transcript should be in the node. Was 'insituNet' applied to all the section or just the granuloma? On top of the panel network the authors can state what region was used to perform 'insituNet' (similar comment for Fig 4A).
- Fig.2C: the color code refer to Fig.2B? The link between the panel should be made more clear. Also Fig.3D: I would advice to integrate the color code directly on the figure.
- Fig5A-B: the color code should here again better explained. 'U' should be defined in the figure.

Minor points

- S-fig2 legend: panel are wrongly annotated
- S-fig2E : give the color code

- Gene names should be written with more consistency: 'cd3' vs 'cd3e', 'inos' or 'inos2', why actb is written 'ACTB', italic or not (S-fig2E'), etc...

Response to reviewers

Dear Sirs,

Thank you for your time and effort for reviewing our manuscript and the positive criticism. Please find pasted below the comments raised and our responses to these point by point in a separate document (in blue). We have indicated when modifications to the manuscript were made.

Remarks to the authors:

Reviewer 1.

Carow et al. report on temporal localization of immune transcripts in the tuberculosis granuloma. They performed simultaneous in situ sequencing of 34 immune transcripts in lung sections from *Mycobacterium tuberculosis*-infected and C57BL/6 C3HeB/FeJ mice, the latter containing both encapsulated and non encapsulated granulomas. The RNAs analyzed are selected to represent various cell types expected in the granulomas. The rolling circle in situ amplification appears to be sufficiently robust to align specific RNA transcripts with cell type and subcellular proximity to *M.tb*. With extensive imaging, this study provides a detailed view of the cellular structure of granulomas, inferred from transcript profiles, providing new information about the time course of granuloma maturation, transcript compartmentalization, and heterogeneity. For example, the RNA expression landscape shows that encapsulated granulomas have necrotic centers with immunosuppressive transcripts and those in

rims associated with activated T cells and macrophages. Likely representing activated infected macrophages, *cd68*, *cd11b*, *tnf* and *socs3* mRNAs were enriched in close proximity to *M.tb*. On the other hand, *ccr6*, *cd19*, *il12*, *cd3*, *cxcr3* and *cd8b* mRNA are enriched in lymphoid over epithelioid areas. These are all interesting observations that provide new insights, with good discussion how the data fit into the abundant literature on granuloma structure and function.

In the discussion, the authors cite a study using a combination of laser microdissection and mass spectrometry to analyze the abundance of proteins in

granulomas regions – maybe this study could have been better represented in the Introduction, to highlight what new information is expected from the RNA profiles.

Marakalala, M. J. et al (1) used laser microdissection and non-targeted proteomic (LC-MS) analysis of the Tuberculosis granuloma in patients, comparing samples obtained from different regions in the granuloma (rim and center for example). LC-MS used in this study did not provide cellular resolution, which instead is achieved by *in situ* sequencing.

On the other hand, transcripts analyzed by *in situ* sequencing are targeted while LC-MS analysis is non-targeted. To our knowledge this is first study in which the molecular composition is studied at a cellular resolution in the context of tissue morphology. Such resolution allows the definition of networks of co-localizing transcripts. This can be used to envisage different mechanisms of protection/pathogenesis that cannot be unraveled by a bulk analysis of granuloma areas. We indicate in the revised version that *in situ* sequencing or other spatially resolved proteomic and transcriptomic approaches could complement and validate each other.

As discussed below, the *in situ* sequencing provides estimates of the relative density of different mRNAs in different locations, while absolute quantity the different mRNAs cannot be measured since the performance of the different padlock probes is uneven. While *in situ* sequencing assay can in principle detect mRNA at low densities, the detection of proteins by MS- is limited by their concentration in the sample. We have highlighted this in the discussion. Given these differences we prefer not to include the MS- information in the introduction section, but have extended the discussion. We have also indicated the ability of LC-MS study to discriminate signals that are physically segregated within each granuloma.

While the study is largely descriptive, the authors propose that "each granuloma probably represents a limited microenvironment that might be influenced independently from the others .. by the quality of the local immune response and the level of inflammation", "Our findings emphasize the importance of analyses at the

tissue-level rather than in blood cell suspensions or in tissue homogenates”, and “the lymphoid-rich areas may constitute the structure where host immune responses to *M. tuberculosis* are orchestrated. This has implications for the design of mucosal vaccines.” These are rather sweeping statements that lack some granular understanding of what exactly is new, and what are the implications for vaccine design?

Wolfgang Sadec

We appreciated this comment and agree not to overstate our findings. We have erased the paragraphs “Our findings emphasize the importance of analyses at the tissue-level rather than in blood cell suspensions or in tissue homogenates from tissues for an accurate link to clinical disease and for a selection of proper determinants of disease progression “ (page 19) and “this has implications for vaccine design” (page 14).

We rephrased the hypothesis paragraph: “each granuloma probably represents a limited microenvironment that might be influenced independently from the others .. by the quality of the local immune response and the level of inflammation” to “Thus, heterogeneity of transcript networks even between histologically similar granulomas may represent a limited microenvironment that might be influenced independently from the others in the same lung by the quality of the local immune response and the level of inflammation”. Even with regard of our limited study size, we feel that this suggestion is legit in the context of our results and discussion.

We have also added a succinct paragraph at the end of the discussion section of the revised manuscript summarizing the novel information from our study “In summary, granulomas showed increasing complexity of co-expressing molecular networks with time after infection. Such increased complexity was due to the presence of adaptive immune transcripts, some of which co-expressed in lymphoid clusters. *M. tuberculosis* co-localized with transcripts from activated macrophages in non-organized granulomas. Encapsulated granulomas showed necrotic centers with

transcripts associated with anti-inflammatory responses. Instead, those in the outer rim of necrotic granulomas or in cellular non-organized lesions in the same lung showed higher abundance of pro-inflammatory networks. Morphologically similar lesions showed highly diverse transcripts networks with common cores.” (page 17).

Reviewer #2 (Remarks to the Author):

Work by Carow, Hauling et al utilized in situ sequencing of 34 transcripts to define spatio-temporal immune landscapes in lung sections from Mtb-infected mice. They show that transcript clusters do not differ much by density earlier on during the infection, but that there is more compartmentalization of transcript densities within granulomas at later time points after infection. Using encapsulated mice granulomas, the authors also show that granuloma centers are enriched with transcripts associated with immune suppression whereas cellular granulomas are enriched with transcripts associated with protective responses. The work is novel in the use of a highly sensitive in situ imaging of mRNA species to characterize time-specific immune diversity in TB granulomas. Their findings on spatial segregation of various immune factors is complementary to and extends results on human and non human primate TB granuloma heterogeneity/variability recently published using PET imaging, flow cytometry transcriptomics and proteomics. The impressive technology applied here is a major strength of the manuscript, and it will definitely be of great interest in the community and to the wider field.

All in all, the manuscript is well written and presented.

Below are the issues that can be addressed to improve the manuscript:

Major comments:

- What is the basis on which the 34 transcripts were chosen for analysis in this study? Is there a rationale why this specific list was considered?

The immune transcripts selected coding for chemokine receptors, cytokines, transcription factors, effector molecules and surface molecules that define innate and adaptive immune populations. Macrophages, B cells and monocyte/ macrophages and neutrophils were defined by these probes. Different subpopulations of T cells were characterized as well. The cells or immune factors identified targeted have been shown to play a relevant role in the cellular immune responses controlling *M. tuberculosis* infection. As controls included *Cc10* mRNA expressed by Clara cells in the airway epithelia, *Spc* mRNA coding for the surfactant protein C expressed in the alveolar epithelial cells and *Actb* mRNA coding for β -actin.

We have now added that “The *in situ* sequencing technique was used to localize simultaneously 34 immune in paraformaldehyde-fixed sections of lungs from *M. tuberculosis*-infected mice. Transcripts targeted code for chemokine receptors, cytokines, effector molecules, transcription factors and surface molecules that define immune cell populations. These molecules play a major role in the cellular immune control of the *M. tuberculosis*-infection” (page 5, Results section).

Throughout the manuscript, the authors show image analysis of one lung tissue sample at each time point post infection (3, 8, 12 weeks, Kramnik). Are these a representation of more lung samples analyzed at these time points? If so, are the image analyses reproducible in the other lung samples?

A limitation of our study is the small number of independent sections examined. This is due to the extensive image acquisition and data processing. In the discussion section we have indicated that the results presented were confirmed in one independent sample for each condition. The reproducibility of the determinations presented were confirmed in three consecutive sections for each condition (page 17, line 17-22). We show in this letter the independent samples analysed:

The lung section from the additional **3 week time point after infection** displayed a similar histopathology compared to the own shown in manuscript (**Figure 1A**). Plotting of signals showed an early localization of *Inos*, *Tnf* and *Cd68*

mRNA in lesions, whereas *Tcrb* was spread out over the tissue similar to findings in manuscript (*Figure 1D*, manuscript Figure 1). A network analysis showed the same core network and similar common interactions as those presented in the manuscript (*Figure 1B, 1C* and manuscript Figure 2A, Figure 6C).

Figure 1. Analysis of the independent sample at 3 weeks after infection

The analysis of lymphoid areas in the additional **8 week time point after infection** showed a similar pattern of transcript localization and interactions with *Cd19* mRNA as the highest connected node (*Fig 2A-C*) and manuscript Fig 3A, 4A, 6A). It also confirmed the heterogeneity between different lymphoid regions that are based on common and unique interactions (*Figure 2B*). The presented networks in the manuscript Figure 4A are less complex as interactions took place in at least 2

consecutive sections whereas in the independent sample interactions of one section of the analysed region are included.

The co-localization of *Cd68*, *Inos* and *Tnf*, the more limited distribution of *Tcrb* to the granuloma and the focal expression of *Cd19* mRNA (Figure 2D) are

similar to those shown in the manuscript Figures 1, 3 and 4.

Figure 2. Analysis of the independent sample at 8 weeks after infection

The additional **12 week time point** the lung showed severe coalescent lesions covering a large fraction of the area. Lymphoid structures with a localized

dense *Cd19* mRNA expression showed no-colocalization with surrounding *Cd68* and *Inos* mRNA, as described in the manuscript (**Figure 3** and manuscript Figure 3B). These results were not added to the manuscript but we indicated that results shown were also observed in an independent sample.

Figure 3. Localization of *Cd68*, *Inos* and *Cd19* in an independent lung sample at 12 weeks after *M. tuberculosis* infection.

- It would be interesting to evaluate the performance of the in situ sequencing in heterogeneous lesion types (rather than just one lung section in fig 7) produced in the C3HeB/FeJ mice. Considering that previous work has shown the presence of multiple lesion types with vastly different microenvironments in C3HeB/FeJ mice following aerosol infection with *Mtb* (Ervin et al, *Disease Models & Mechanisms* (2015) 8, 591-602).

Following the suggestion of the reviewer we have now increased the lung area of the C3HeB/FeJ tissue section analyzed by *in situ* sequencing. This has allowed statistical comparison of the data. The HE of the augmented analysis is shown in Supplementary Figure 1 in the revised manuscript, and **Figure 4**. In the

analyzed regions have annotated 7 discrete necrotic encapsulated granulomas (instead of two), 5 non-organized granulomas (in which the center and the rim both annotated) and 2 small granulomas and 3 unaffected regions (*Figure 4*).

Figure 4. H& E staining and areal annotation of the C3HeB/FeJ M. tuberculosis infected tissue analysed.

The PCA analysis includes now 27 samples (*Figure 5* and manuscript *Figure 7E*). We have now added on the PCA analysis prediction ellipses, a “graphical multivariate ANOVA test”. The predictive ellipses displayed have a 95% probability that a new observation from the same group will fall inside the ellipse. The different ellipse center (the data mean) indicate that the data can be statistically segregated, while the rather thin ellipses indicate a good correlation between the

data. Both the PCA analysis and the prediction ellipses were generated with ClustVis.

Figure 5. PCA analysis and prediction ellipses of 27 annotated regions in the C3HeB/FeJ *M. tuberculosis*-infected lung

This new data was also incorporated in Figure 7F and G in the revised manuscript, in which the relative transcript density in different areas in the C3HeB/FeJ granuloma are compared.

In figure 3C, the authors show that cd3 mRNA co-localizes with cd68 mRNA at the epithelioid areas. This localization observation should be strengthened by immunohistochemical validation using antibodies/markers for epithelioid cells. This could be done the same way as in figure 3b, where the authors beautifully confirmed specificity of the in situ sequencing (for cd19 mRNA) by using immunohistochemistry staining of B cells.

Following the suggestion from the reviewer we have performed double immunolabelling of CD68, CD3 and stained with DAPI slides from C57Bl/6 mice

12 weeks after infection with *M. tuberculosis*. The localization of these proteins in two different areas is similar to that of transcripts shown by *in situ* sequencing (manuscript Figure 3C). CD68 was excluded from the DAPI-dense region while CD3 stains both the DAPI-dense and -lighter areas of the lesion (**Figure 6**).

The CD3 and CD68 and secondary antibodies used are included in the Material and Methods section (immunohistochemistry) of the revised manuscript and one micrograph was added as a Supplementary Figure 3B.

Figure 6. Expression of CD3 and CD68 in DAPI dense and surrounding areas of a TB granuloma

More specific details are required to allow better understanding of the methods as well as the ability for a researcher to reproduce the work. For example; what is the number of the lung tissue samples analyzed for each time point, how many mice were used in the experimentation (at each time point), what Mtb strain was used for the aerosol infection?

As indicated above and in the Discussion section of the manuscript, the results presented were generated from the analysis of one animal and were reproduced in one independent sample per condition. The accuracy of determinations was confirmed in 3 consecutive sections for each condition. Results

of the independent sample are shown in this letter, while the information obtained from consecutive sections is indicated throughout in our manuscript. We considered this limitation, due to extensive image and data analysis, in the discussion section. Statistics were performed by comparing frequencies of transcripts in different areas of the lung or in different regions of the granuloma but also by comparison of determinations the consecutive sections, which confirmed the specificity of the signals.

Mice were infected with 200 *M. tuberculosis* Harlingen strain.

- It would be interesting to evaluate if any of the transcripts compartmentalized at later time points would influence disease outcome. The authors should discuss more on potential clinical implications of these findings in the context of TB disease control/progression.

Here a spatially resolved molecular mapping of the TB granuloma is presented. By showing that at different stages of infection and in different types of granuloma the presence of defined transcript networks, and a clear segregation of inflammatory and anti-inflammatory pathways in precise anatomical localizations in the granuloma different hypothesis are presented. The anatomical localization of these networks probably modulates the protective or pathological responses during the infection. A summary of the results obtained and the implication was added in the discussion of the revised manuscript (page 18).

- Also, considering differences between mouse and human TB granulomas, the authors should discuss if the temporal and spatial transcripts observed in this study can be extrapolated for translational interventions in TB patients.

Organized, encapsulated and non-organized lesions (areas of pneumonia and perivascular peribronchiolar infiltrations) and the presence of lymphoid follicles are observed during active TB in humans. It is important to stress that relevant findings

in mechanisms of mouse immune control of *M. tuberculosis* infection have been reproduced in human patients. Thus, the *in situ* sequencing results obtained in the mouse experimental model, despite absence of a latency phase, can be used for further comparison in other animal experimental models of infection (i.e. non-human primates) and for testing the expression of the observed patterns in human granulomas. The method could also be used for comparison of granulomas from latent or vaccinated individuals with those with active TB, and to further understand the many stages of granuloma maturation in humans.

Minor comments:

- Please put scales of the lung images utilized throughout the manuscript.

Scales were added in the lung images throughout the manuscript.

- Numbering on the legends for Figure S2 needs to be fixed.

The numbering of the legends of Supplementary Figure 2 was corrected.

- Under ‘maturation of the granuloma’, authors state that ‘all but 4 mRNA species (cd4, cd40l, elane and il17a) showed increased localization’. I can’t locate CD40L and elane from the density plots in figure 1 and supplemental figure S2e.

All probes did not perform similarly well. The *Cd4*, *Cd40l*, *Elane* and *Il17a* mRNA signals were low and excluded from analysis in several determinations. We have followed throughout the exclusion criteria that if any mRNA species was undetectable in more than two areas in an analyzed section, it was excluded from analysis. This exclusion criteria is added into the legend to Supplementary Figure 2 and in the Results section (page 5). In Figure 1B, *Cd40l* and *Elane* mRNA were excluded following this criterion.

- Under ‘maturation of the granuloma’, authors list several transcripts expressed by myeloid cells at a similar frequency, and then increased enrichment for transcripts that are associated with adaptive immune responses. Among the list of transcripts

associated with adaptive immune responses, the authors include CD68. This molecule is actually a myeloid-specific marker and more abundant in macrophages and monocytes, and not adaptive immune response (<https://www.ncbi.nlm.nih.gov/pubmed/?term=CD68+Chistiakov+DA>).

We appreciate this concern and have moderated the interpretation of our observations. For example: “Instead, increased relative enrichment was observed for several transcripts, **some of which** associated with adaptive immune responses at 12 wpi (i.e. *Cd19*, *Ccr6*, and *Cd3e* mRNA) (Figure 1B)” in page 6 of the revised manuscript. An unpaired Student’s t test was used for comparison of the relative densities of the transcripts in granulomas at 3 and 12 wpi (Figure 1B).

Figure 5b: According the heatmap clustering tree at 12 wpi, ‘U’ and ‘3’ appear to be clustered together; while the actual expression intensities are opposite (3 is entirely red and U is entirely blue). This needs to be reconciled?

Thank you for this comment: The heat map was generated using the web tool ClustVis. The heat map uses linear expression data and is **row-centered**, and unit variance scaling (the standard deviation of measurements in rows was used as the scaling factor). Since only a limited number of samples are shown, the program performed no clustering of the columns. Rows were instead clustered by correlation. We have corrected the heat map and deleted column clustering in the revised version. We have also added an extended the explanation on how data was processed for the heat map analysis in the Material and Methods section (page 22 of the revised manuscript).

Figure 7 legend seems incomplete

We have added the missing information to the legend.

Reviewer #3 (Remarks to the Author):

In this manuscript, Carow et al aim to resolve granuloma spatial structure and uncover granuloma heterogeneity in tissues by combining histology staining and in situ sequencing - a technology able to image RNA transcript localizations on fixed sections. Here 34 transcripts have been simultaneously monitored and a tailored bioinformatics tool (coined 'insituNet') has been used to decipher the transcript co-localization network. After infection with a Mycobacterium tuberculosis, the immune system react by forming multi-cellular structures called granuloma that aims to contain the bacteria. These structures are highly diverse and in a same section, granuloma can be highly different and the method introduced here by the authors is one of the most advance to perform spatial transcriptomics. The application of in situ sequencing to granuloma structure is highly relevant and timely to answer the question of heterogeneity in host-pathogen interaction.

The paper is well written and the analysis is conducted in a clear sequential manner. The analysis is greatly done. Although impressive from a methodological point of view, I feel that the biological outcome of the study is not clearly explained and need to be reinforced. The way that data are displayed can clearly be improved.

- The rational of the selection of the 34 tracked transcripts could be better explained. For example: what genes refer to "macrophage activation"?

The immune transcripts selected coding for chemokine receptors, cytokines, transcription factors, effector molecules and surface molecules that define innate and adaptive immune populations. Macrophages, B cells and monocyte/ macrophages

and neutrophils were defined by these probes. Different subpopulations of T cells were characterized as well. The cells or immune factors identified targeted have been shown to play a relevant role in the cellular immune responses controlling *M. tuberculosis* infection. *Inos* and *Tnf* were associated to macrophage activation. As controls included *Cc10* expressed by Clara cells in the airway epithelia, *Spc*, coding for the surfactant protein C expressed in the alveolar epithelial cells and *Actb* coding for β -actin.

We have now added that “Transcripts targeted code for chemokine receptors, cytokines, effector molecules, transcription factors and surface molecules that define immune cell populations. These molecules play a major role in the cellular immune control of the *M. tuberculosis*-infection.”. (page 5, result section).

- In order to quantify the specific transcripts in the granuloma region the authors several metrics: “fold increase over unaffected area” (s-fig2), Density plots (fig1A), “log₂ mean fold Granuloma/unaffected area” (fig1B), ‘: this is overall very confusing.

In many graphs we have compared the log₂ of the ratios of the **densities** of a defined mRNA species in different areas of the same lung (i.e. granuloma / unaffected, lymphoid/ epithelioid, signals located at less than a certain distance from *M. tuberculosis* etc). To increase consistency in y axis title, we have in these graphics changed the title for “log₂ of x/ y area densities“. We have also corrected original graphs where a log₂ scaling of linear data was instead shown. We hope the interpretation becomes less cumbersome and more regular.

We added density plots as additional visualization of plotted absolute signals as, in some cases, differences in expression were easier to grasp if a 2log signal density estimation.

Overall page 6 line 2-14: this part is cryptic. *Actb* could be added in Fig.1B.

We have rephrased the paragraph in page 6 lines 2-14 in the revised version. *Actb* mRNA relative densities are included in figure 1B. In all graphs we have added *Actb* as the last mRNA species analysed.

Why Fig 1B does not contain the ‘8 weeks’ time point?

We wanted to exemplify the differences at an early and late time point after infection, these differences were difficult to highlight when the 8-week time point was included (that resembled the 12 week time point).

Other example: is really cd68 above actb level?

The relative densities of *Cd68 mRNA* in Figure 1B are similar to those of *Actb*.

In order to clarify: Since the padlock probes used for *in situ* sequencing do not show equal performance, we are not able to compare the densities of two different mRNAs in a certain location or obtain absolute quantification of a certain mRNA species (as can be done with real time PCR where the amplification performance is assumed to be 100%). However, the performance efficiency for each probe is highly reproducible, which means that relative quantification can be efficiently done.

All over we compared the relative densities of the same probe in different areas of the section. Thus, we cannot indicate that *Cd68* mRNA has a higher density than *Actb* mRNA. Instead, we conclude that the **relative density** of *Cd68* mRNA in a lymphoid compared to epithelioid area is lower than the relative density of *Actb* in the same region (Figure 3D). We reason that then CD68+ cells preferentially locate in the epithelioid region of the granuloma (as expected).

The specificity, reproducibility and performance were described in Supplementary Figure 2 and page 5 of the results section.

- Transcript co-expression is not clear for me.

The transcription co-expression was studied by InsituNet, as explained in the methods sections. In short, the program analyses for each possible transcript pair that their occurrence at $\leq 10 \mu\text{m}$ in an annotated area is significantly higher than expected considering their occurrence on the whole section. An example of this analysis is provide in Supplementary Table S1, where statistically significant sequence pairs (χ^2 test) at $< 10 \mu\text{m}$ in a lymphoid-rich region. A network of co-localizing transcripts is constructed from this data.

Why *actb* is not in? Just by abundance the transcript should be in the node.

Generally, a high expression does not automatically lead to interactions. The program InsituNet takes the abundancy in consideration and only if co-expression occur at higher frequency than expected based on the mRNAs' abundances in the whole section it will be displayed as an interaction. For example, our highest detected *Cc10* mRNA, that is not an immune transcript, and that localizes with airways did not display any interactions.

Anyhow, since *Actb* mRNA is expressed in all cells and thus might disturb the network visualization of immune markers, we have removed it from the network visualization in cases when it was present. We have added this information in the legend to Figure 2 (page 33): "*Actb* mRNA was excluded from network analysis due to broad expression in different cellular populations".

Was 'insituNet' applied to all the section or just the granuloma? On top of the panel network the authors can state what region was used to perform 'insituNet' (similar comment for Fig 4A).

The co-expression analysis was done for the whole granuloma in figure 2.

The networks of colocalizing transcripts shown were calculated from a single granuloma at the indicated time points after infection in Figure 2 of our manuscript. We identified the same granuloma in all three consecutive sections and display interactions found in at least 2 of the 3 slides. We have indicated in the legend to

figure 2: “Representative examples of one lesion per time point annotated in Supplementary Figure 1 were selected. The significantly co-expressed sequences that are common in the same lesion from at least two consecutive sections are here depicted”.

- Fig.2C: the color code refer to Fig.2B? The link between the panel should be made more clear. Also Fig.3D: I would advice to integrate the color code directly on the figure.

Thank you for this suggestion. We have indicated in the legend to the figure that the centroid analysis follows the same color codes than the hexbins. We added in the legend to Fig2 B “The color-code used for the bars corresponds to that in the 2D-hexbin map”, and connected with arrows the centroid analysis graphs and the hexbin maps.

The color code in Figure 3D was integrated in the figure.

- Fig5A-B: the color code should here again better explained. ‘U’ should be defined in the figure.

The heat map was visualized using the web tool ClustVis. The heat map uses normalized linear expression data and is **row-centered**. Unit variance scaling (the standard deviation was used as the scaling factor) was performed for rows. Further explanation was added into the material and methods section.

U: unaffected area was defined in the figure (Unaff.)

Minor points

- S-fig2 legend: panel are wrongly annotated

We have corrected this mistake.

- S-fig2E : give the color code

The color code is now indicated in the figure in the revised version.

- Gene names should be written with more consistency: ‘cd3’ vs ‘cd3e’, ‘inos’ or ‘inos2’, why actb is written ‘ACTB’, italic or not (S-fig2E’), etc...

Thanks for pointing this out. We have corrected this mistake. We have consistently written the transcripts in italics, capitalizing each one in all graphs we could change. Note that a part of the data could not be modified (network analysis based on decoded csv files) as our initial decoding legend had iNos2 and ACTB in it.

We appreciate the time and effort dedicated by the reviewers, and hope our response addresses their suggestions and concerns on our manuscript.

Sincerely ,

REVIEWERS' COMMENTS:

Reviewer #2 (Remarks to the Author):

This reviewer is satisfied with the author responses and extra work done to improve the manuscript. This work will definitely be of great interest in the TB community and to the wider field.

Reviewer #3 (Remarks to the Author):

All the points I have raised are addressed in the revision and I have no further comments.